# Structural analysis of human CEACAM1 oligomerization

Amit K. Gandhi [1,5✉], Zhen-Yu J. Sun [2,5], Yu-Hwa Huang[1], Walter M. Kim[1], Chao Yang[1], Gregory A. Petsko [3], Nicole Beauchemin[4] & Richard S. Blumberg [1✉]

The human (h) CEACAM1 GFCC' face serves as a binding site for homophilic and heterophilic interactions with various microbial and host ligands. hCEACAM1 has also been observed to form oligomers and micro-clusters on the cell surface which are thought to regulate hCEACAM1-mediated signaling. However, the structural basis for hCEACAM1 higher-order oligomerization is currently unknown. To understand this, we report a hCEACAM1 IgV oligomer crystal structure which shows how GFCC' face-mediated homodimerization enables highly flexible ABED face interactions to arise. Structural modeling and nuclear magnetic resonance (NMR) studies predict that such oligomerization is not impeded by the presence of carbohydrate side-chain modifications. In addition, using UV spectroscopy and NMR studies, we show that oligomerization is further facilitated by the presence of a conserved metal ion ($Zn^{++}$ or $Ni^{++}$) binding site on the G strand of the FG loop. Together these studies provide biophysical insights on how GFCC' and ABED face interactions together with metal ion binding may facilitate hCEACAM1 oligomerization beyond dimerization.

[1] Division of Gastroenterology, Department of Medicine, Brigham and Women's Hospital, Harvard Medical School, 75 Francis Street, Boston, MA 02115, USA. [2] Department of Cancer Biology, Dana-Farber Cancer Institute, Boston, MA 02215, USA. [3] Ann Romney Center for Neurologic Diseases, Department of Neurology, Brigham and Women's Hospital, Harvard Medical School, Boston, MA 02115, USA. [4] Rosalind and Morris Goodman Cancer Institute, McGill University, Montreal, QC, Canada. [5]These authors contributed equally: Amit K. Gandhi, Zhen-Yu J. Sun. ✉email: agandhi2@bwh.harvard.edu; rblumberg@bwh.harvard.edu

Human (h) carcinoembryonic antigen-related cell adhesion molecule 1 (CEACAM1) is a type I glycosylated immunoglobulin (Ig)-related transmembrane protein that plays an important role in physiological processes associated with a variety of hematopoietic and stromal populations (reviewed in 1)[1–5]. hCEACAM1 is characterized by 12 structural isoforms that are generated by alternative splicing and include a common N-terminal immunoglobulin-like (IgV) domain, variable numbers of immunoglobulin (IgC2)-like domains and a transmembrane region that is coupled to either a short (S) or long (L) cytoplasmic tail important for signaling and inhibitory function in the case of L-isoforms which contain immunoreceptor tyrosine-based inhibitory motifs[2,6]. In addition, IgV-like N-domain containing secreted isoforms also exist[2]. Like other hCEACAM family members, the hCEACAM1 IgV domain serves as the major ligand binding surface and is characterized by the presence of two anti-parallel β-sheets creating two distinct faces, the GFCC′ and ABED faces, respectively[7–9].

The major mode of hCEACAM1 binding is homophilic such that it exists on the cell surface as collections of monomers and cis-dimers. This monomer:dimer equilibrium is mainly determined by interactions between the GFCC′ faces of the IgV domains of opposing hCEACAM1 molecules, residues within the transmembrane domain and intracellular calcium concentrations mediated by cellular activation where increased calcium levels promotes hCEACAM1 monomerization[1,2,7,10]. The formation of cell surface hCEACAM1 monomers allows for trans-homophilic or heterophilic interactions critical to the induction of biologic responses[1,2,7,10]. Heterophilic ligands include a variety of microbes and host cell proteins such as hCEACAM5, hCEA-CAM6, the T cell inhibitory and mucin domain containing protein 3 (TIM-3) and potentially programmed cell death protein 1 (PD-1)[9–21]. In each case where it has been biochemically or biophysically defined, homophilic and heterophilic ligation depends upon GFCC′ face interactions[9–21]. Consistent with the importance of the GFCC′ face for ligand interactions, the GFCC′ face also governs various states involved in hCEACAM1 dimerization. As such, size-exclusion chromatography with multi-angle light scattering (SEC-MALS) studies of hCEACAM1 Ig-V and GFCC′-face mutants (V39A, I91A, N97A, E99A) detected various states of hCEACAM1 including monomeric, transition and dimeric states[7].

hCEACAM1 has also been shown to form higher-order homo-oligomers and micro-clusters which, when present in a trans-configuration have been hypothesized to be important for strength of signal transduction[22–24]. In support of hCEACAM1 oligomerization, cross-linking, and low resolution (20 Å) molecular tomography studies of liposomal-immobilized rat CEA-CAM1 (IgV and 3 IgC2 domains), sharing ~43% sequence identity with hCEACAM1 in the IgV domain, revealed multiple states of CEACAM1 that included monomers, dimers, trimers and micro-clusters of closely associated molecules[23]. CEACAM1 higher-order clustering is also supported by studies by Grasberger et al. in other systems where it was shown that the propensity for trimer and higher-order oligomer formation significantly increases when proteins with adhesive interfaces are oriented on membrane surfaces[25].

The structural basis for hCEACAM1 homo-oligomerization and the formation of higher-order structures is unknown. It is currently thought to be initiated by IgV GFCC′ face binding and involve other interactions that facilitate CEACAM1 packing as multimers[7]. Although the GFCC′ face interactions are well-characterized and shown to be dependent upon a conserved set of amino acids during homophilic or heterophilic binding to a wide variety of ligands[2,7,9], the other structural determinants potentially involved in multimeric interactions are poorly characterized. Interestingly, a previous x-ray crystal structure of wild-type (WT) hCEACAM1 IgV (PDB code 2GK2) showed three hydrogen-bond interactions through the ABED faces of two hCEACAM1 molecules involving residue N70 of each chain with residues Glu (E in single-letter amino acid code)16 and Tyr(Y)68 in addition to a hydrogen-bond interaction between the Y68 and Ser(S)72 residues (Supplementary Table 1)[8]. Similar ABED face-associated, hydrogen-bond interactions have also been observed in a hCEACAM1 IgV mutant crystal containing an alanine (A) substitution at the Val(V)39 residue in the hCEACAM1 GFCC′ face that allows the majority of GFCC′-mediated interactions to occur before entering a transition state (PDB code 6XNW). However, a hCEACAM1 IgV mutant that contains a critical mutation at the Asn(N)97 residue that disables GFCC′-mediated binding (PDB code 6XO1) does not show similar ABED face-association; wherein, a single hydrogen-bond interaction is observed between residues Gln(Q)26 and Ile(I)67[7]. These studies suggest that the optimal formation of ABED interactions is dependent upon GFCC′-mediated homodimerization. However, as these structural studies were performed with non-glycosylated hCEACAM1 IgV proteins and the N70 residue is contained within an N-linked carbohydrate side-chain consensus motif, a steric clash might occur for glycosylated hCEACAM1; thus the physiological relevance of these observed ABED face interactions remains unclear.

Another interesting observation that provides insights into hCEACAM1 oligomerization is the discovery that the crystal structures of WT hCEACAM1 (PDB code 2GK2) and a V39A mutant of hCEACAM1 (PDB code 6XNW) IgV domains were resolved with bound nickel (Ni$^{++}$), which was shown to bridge three molecules of hCEACAM1[7,8]. In these crystal structures, Ni$^{++}$ was involved in hexadentate interactions with three His(H) 105 side-chains and three carbonyl groups of V106 residues that are contained within the G strands of a hCEACAM1 molecule and its two symmetry mates. These studies further suggest that metal binding to residues within the G strand of the FG loop may potentially be involved in hCEACAM1 multimerization and/or its stabilization once formed.

Here, we sought to understand the structural basis for the propagation of hCEACAM1 oligomers through the IgV GFCC′ and ABED faces, the potential impact of glycosylation sites contained within the IgV domain and the role played by metal ions. Specifically, we report the crystal structure of a hCEACAM1 oligomer at an atomic resolution level (2.2 Å). This crystal structure provides evidence for the simultaneous presence of two adhesion sites on a single hCEACAM1 molecule that involves the ABED and GFCC′ faces of the IgV domain. Further, NMR spectroscopy along with the modeling of sugar molecules on the crystal structure allow us to predict that such oligomers can accommodate glycosylation without steric hindrance. In addition, UV spectroscopy and NMR studies in solution revealed that Ni$^{++}$ and Zn$^{++}$ binding with hCEACAM1 mediates an equilibrium between hCEACAM1 dimers and higher-order oligomers. Overall, these studies provide a structural basis for hCEACAM1 clustering and higher-order oligomerization that involves both the GFCC′ and ABED face of the IgV domain and shows that dynamic metal ion bridging of hCEACAM1 molecules could be important for its cellular functions and signaling.

## Results

### Crystal structure of hCEACAM1 oligomer with two adhesion sites and simultaneous interactions through GFCC′ and ABED face.

To understand the structural basis for higher-order oligomerization and clustering of hCEACAM1, we solved the crystal structure of a tagless hCEACAM1 IgV domain at 2.2 Å resolution (Fig. 1a–c and Supplementary Figs. 1, 2). Crystallographic data

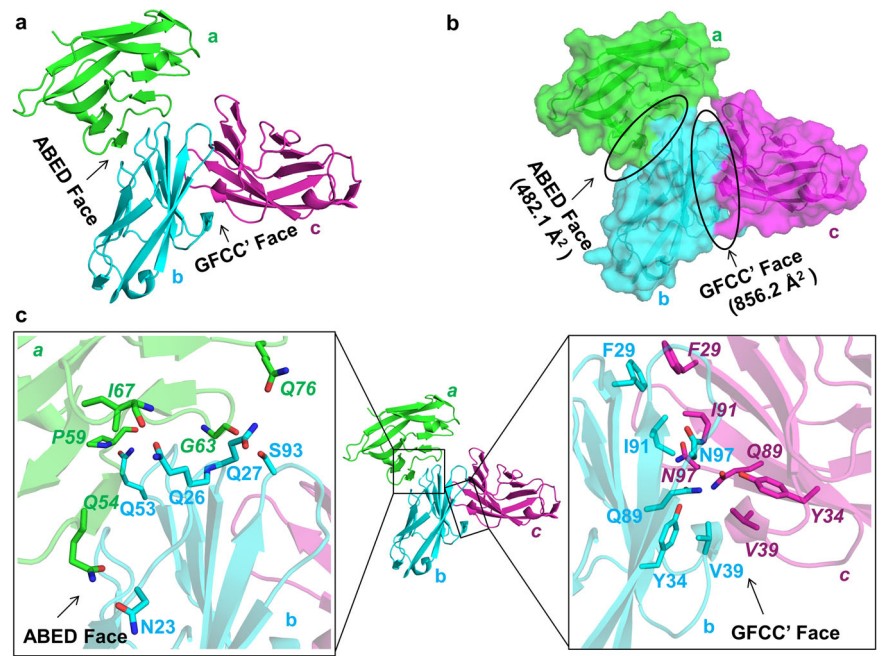

**Fig. 1 Crystal structure of a hCEACAM1 oligomer (PDB code 7RPP resolution (2.20 Å) involving GFCC' and ABED interfaces. a** Ribbon diagram of the hCEACAM1 IgV domain, whereas hCEACAM1 molecule b (cyan) and molecule c (magenta) make GFCC' face interactions and molecule a (green) and molecule b (cyan) make ABED face interactions. **b** Surface (green, cyan and magenta) representation for molecule a, b, and c respectively, which are involved in simultaneous GFCC' (856.2 Å²) and ABED (482.1 Å²) face interactions (shown by black oblong). **c** Ribbon diagram of the hCEACAM1 IgV domain with GFCC' and ABED face interactions. The left inset shows ABED face interactions between residues of molecules a (green) and molecule b (cyan), wherein residues Q54, P59, G63 I67, Q76 of molecule *a* (green) make six hydrogen-bond interactions with residues N23, Q26, Q27, Q53, S93 of molecule b (cyan). The carbon atoms in green (molecule a) or cyan (molecule b), carbonyl oxygen in red and nitrogen in blue, are colored, respectively. The right inset shows GFCC' face interactions between residues of molecules b (cyan) and molecule c (magenta), wherein residues F29, Y34, V39, Q89, I91, and N97 of molecules b and c make various hydrogen-bond and hydrophobic interactions with each other. The carbon atoms in cyan (molecule b) or magenta (molecule c), carbonyl oxygen in red and nitrogen in blue, are colored, respectively. The residues of molecule a and molecule c are annotated in italics in left and right insets, respectively.

**Table 1 Data collection and refinement statistics (molecular replacement).**

| Human CEACAM1 Oligomer (PDB code 7RPP) | |
|---|---|
| **Data Collection** | |
| Space group | C121 |
| Cell dimensions | |
| a, b, c (Å) | 73.13, 86.06, 66.08 |
| α, β, γ (°) | 90.0, 100.27, 90.0 |
| Resolution (Å) | 29.86–2.20 (2.32–2.20) |
| $R_{merge}$ (%)ʹ | 13.3 (76.7) |
| I/ σI | 7.3 (2.1) |
| Completeness (%) | 100 (100) |
| Redundancy | 4.9 (4.7) |
| **Refinement** | |
| Resolution (Å) | 29.86–2.20 (2.32–2.20) |
| No. reflections[a] | 101,329 (13,993) |
| $R_{work}/R_{free}$ | 18.4/23.7 |
| No. atoms | |
| Protein | 2520 |
| Ligand/ion | 8 |
| Water | 74 |
| *B-factors* | |
| Protein | 33.0 |
| Water | 34.73 |
| *R.m.s.deviations* | |
| Bond-lengths (Å) | 0.016 |
| Bond-angles (°) | 1.78 |

[a]Values in the parentheses are for highest resolution shell.

collection statistics and structural refinement are summarized in Table 1. The hCEACAM1 oligomer crystal structure revealed three hCEACAM1 molecules (a, b, c) in the asymmetric unit (Fig. 1a and Supplementary Fig. 1a–c) that exhibited an overall similarity in their structure based upon evidence of an anti-parallel beta-sandwich fold of all three molecules compared to a single hCEACAM1 molecule as previously described for a hCEACAM1 homodimer structure (PDB code 4QXW)[9]. We found that molecules (b) and (c) formed a dimer through GFCC' face interactions (Supplementary Table 2), which resembled the WT dimer (PDB code 4QXW) with a C-alpha root mean square deviation (RMSD) of 0.66 Å (over 1489 atoms) and an interaction interface area of 856.2 Å² (Fig. 1b), that was slightly larger than the interface area of a previously described homodimer (824.6 Å², PDB code 4QXW)[9]. This GFCC' face interaction of the oligomer was mediated by various hydrogen-bond interactions that included residues Phe(F)29, S32, Y34, V39, Gly(G)41, Gln(Q)44, Gln(Q)89, Ile(I)91, N97 and E99 and hydrophobic interactions involving F29, I91 and V39 residues as previously described for a hCEACAM1 GFCC'-mediated homodimer (Fig. 1c, Supplementary Figs. 1c, 2a, b and Supplementary Table 2).

We also observed changes in the GFCC' interface in association with an oligomer relative to that involved in a homodimer. Specifically, we observed formation of two new hydrogen-bond interactions between residues E37 and Arg(R)38 of both molecules (b & c) in the oligomer that were not evident in the crystal structure of a hCEACAM1 dimer (PDB code 4QXW)[9] (Supplementary Fig. 2c, d). In addition, compared to the asymmetric formation of seven hydrogen-bond interactions of the N97 residue in a hCEACAM1 dimer (PDB code 4QXW), we

observed that the N97 residues of molecule (b) and (c) in the oligomer participated in four symmetrical hydrogen-bond interactions in the formation of the GFCC' interaction face (Supplementary Fig. 2a). This indicates that the GFCC' face exhibits some degree of flexibility in accommodating the formation of a higher-order oligomer such that it's surface area further increased relative to that observed with a homodimer (Supplementary Table 2).

Notably, molecule (b) showed an additional interaction site involving the ABED surface with an interface area of 482.1 Å$^2$ (Fig. 1b) which was mediated by interactions with molecule (a) through residues located at its respective ABED face (Fig. 1a–c and Supplementary Fig. 1a–b, 3a). This involved six hydrogen-bond interactions (Supplementary Table 3) between the residues Q26-I67 (using nomenclature convention here and after, where Q26 residue is from molecule (b) and I67 residue in italics is from molecule (a)), Q27-Q76, Q53-Pro(P)59, S93-G63 and N23-Q54 (Fig. 1c and Supplementary Figs. 1b, 3a). The residues Q26 of molecule (b) and I67 of molecule (a) mediated two hydrogen-bond interactions and formed the central site for the formation of the ABED face, wherein the side-chain atoms NE2 and OE1 of molecule (b) residue Q26 made two hydrogen-bonds of 3.0 Å and 3.1 Å with the main chain oxygen and nitrogen atoms of molecule (a) residue I67, respectively (Fig. 1c and Supplementary Figs. 1b, 3a). Consistent with the centrality of the Q26-I67 interaction in the formation of an ABED-based interface interactions, these two amino acids were the only contact sites observed in a previously described N97A mutant crystal structure (Supplementary Fig. 3b) which otherwise disabled hCEACAM1 IgV GFCC' face-mediated interactions[7]. In addition, molecule (b) residues N23 and Q27 made side-chain to side-chain hydrogen-bond interactions of 2.7 Å and 3.1 Å with molecule (a) residues Q54 and Q76, respectively (Fig. 1c and Supplementary Figs. 1b, 3a). Side-chain to main-chain backbone hydrogen-bonds of 3 Å between molecule (b) residue Q53 and molecule (a) residue P59 further strengthened the ABED face interactions (Supplementary Figs. 1b, 3a). In contrast to the GFCC' interface, the ABED interface did not involve hydrophobic interactions.

Interestingly, molecule (b) residue S93, which is located within the FG loop, made a side-chain to backbone hydrogen-bond interaction of 2.8 Å through molecule (a) residue G63 (Supplementary Fig. 3a). This unique interaction between a GFCC' face and ABED face residue demonstrates the coordination between CEACAM1 residues in the formation of oligomers. Taken together, this oligomer crystal structure revealed simultaneous interactions through a primary (GFCC'-GFCC') and secondary (ABED-ABED) contact at an atomic resolution level thus demonstrating the presence of two adhesion sites on a single hCEACAM1 IgV molecule. Such observations provide a structural foundation for understanding hCEACAM1 oligomerization.

**Glycosylation of residues N70, N77 and N81 sites are not predicted to impede GFCC' and ABED face hCEACAM1 oligomer interactions.** Recent NMR studies have aimed to understand the role of glycosylation in hCEACAM1 GFCC' face-mediated dimerization with inconclusive results[26,27]. To better understand the potential impact of glycosylation in the formation of hCEACAM1 GFCC' and ABED face interactions, we modeled the N-linked sugars N-acetylglucosamine (NAG) and β-d-Mannose (BMA) on the N70, N77 and N81 residues within the ABED face residues of all three human CEACAM1 molecules (a, b, c) in the hCEACAM1 oligomeric crystal structure shown above (Fig. 2a–c). This manual modeling of the first 3 sugars of the native glycans was based on the mouse CEACAM1 crystal structure (PDB code 1L6Z) wherein a conserved N70

glycosylation site residue was resolved and shown to be linked to two molecules of NAG and a molecule of BMA (Supplementary Fig. 4a, b)[28]. Consistent with this, our previous crystal structure of a hCEACAM1 dimer (PDB code 4QXW) supported this modeling as it showed binding of octyl beta-D-glucopyranoside (BOG), a glycan mimic, near to residue N70 (Supplementary Fig. 4c) which overlapped with the N70-linked sugar moieties modeled from mouse CEACAM1 (Supplementary Fig. 4d)[9]. We thus optimized a glycosylated model of hCEACAM1 with N-linked sugars at residues N70, N77 and N81 of each CEACAM1 molecule of our oligomer structure. We observed that none of the modeled sugar molecules exhibited evidence that they caused steric-hindrance to the ABED face hydrogen-bond interactions (Fig. 2a–c). In addition, none of the sugar molecules affected or blocked any of the residues which mediate GFCC' face interactions in the formation of a homodimer based upon this modeling (Fig. 2a–c). Further, as an independent measure of our manual modeling, sugar molecule modeling with the CHARMM glycosylation server[29] showed a similar model of hCEACAM1 glycosylation without any steric clashes at the GFCC' and ABED faces (Supplementary Fig. 5a). Consistent with these structural analyses, a previous study showed glycosylation doesn't affect CEACAM1 GFCC'-face mediated interactions with a microbial ligand[11].

We also performed a nuclear magnetic resonance (NMR) spectroscopy study of $^{15}$N-labeled wild-type hCEACAM1 IgV protein in solution with or without BOG taking advantage of our previously described NMR assignments[7]. This showed that addition of BOG caused no peak shift changes at the GFCC'-based dimer interface (Fig. 3a and Supplementary Fig. 6) and only caused a few minor peak shifts that were localized to residues Leu(L)18 and L73 that were not involved in ABED face interactions (Fig. 3b, c). This was consistent with the observed binding of BOG proximal to residues L18, N70 and L73 (Supplementary Fig. 4c) in a previously described WT (PDB code 4QXW) or I91A IgV mutant (PDB code 6XNT) crystal structures[7,9]. Thus, NMR studies of hCEACAM1 and BOG showed no significant changes at the ABED and GFCC' faces that would anticipate the formation of steric clashing due to glycosylation as predicted by the modeling of sugar molecules onto the hCEACAM1 crystal structure thus allowing for the formation of the oligomeric structure observed.

**Human CEACAM1 GFCC' face-mediates higher-order oligomerization and micro-cluster formation is supported by flexible ABED face interactions.** Previous structural studies of wild-type and GFCC' face mutants of the hCEACAM1 IgV domain and the determination of ligand binding to hCEACAM1 have underscored the importance of the GFCC' face as representing a central force in hCEACAM1 interactions and monomer:dimer equillibirum[7,9]. In addition, the N97A mutation which disables GFCC'-mediated dimerization (PDB code 6XO1) supports the importance of the GFCC' face in oligomerization as well as this mutant does not show proper ABED face-associations and only exhibits minor contacts via hydrogen-bond interactions involving residues Q26 and I67 (Supplementary Fig. 3b)[7]. Consistent with the central role of GFCC' face in dimerization and subsequent oligomerization, we observed GFCC' face mediated interactions between two oligomers that enabled the formation of higher-order oligomers (Fig. 4a), wherein molecule (a) of the oligomer present in the crystal asymmetric unit mediated GFCC' face interactions with a symmetry mate of molecule a (a$_s$) of the symmetry-related second oligomer (Fig. 4a). Superimposition of the oligomer present in the crystal asymmetric unit and symmetry-related second oligomer revealed the GFCC' face

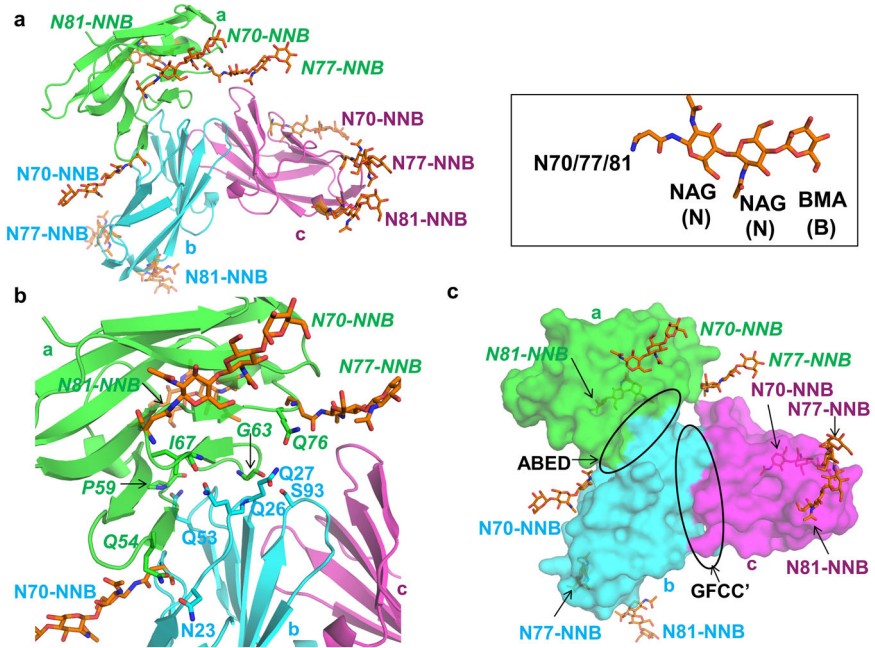

**Fig. 2 Glycosylated model of the hCEACAM1. a** Modeling of the N-linked sugars N-acetylglucosamine (NAG) and β-d-Mannose (BMA) onto the N70, N77, and N81 residues of all three human CEACAM1 molecules observed in the crystal structure, wherein hCEACAM1 molecules (a in green, b in cyan, and c in magenta) are shown by ribbon diagram. Right inset shows structure of two N-acetylglucosamine (NAG) and one β-d-Mannose (BMA) by stick representation, which are modeled onto each set of N70, N77, and N81 residues in the crystal structure shown in left and abbreviated as NNB in the figure. For each sugar molecule, carbon atoms in orange, carbonyl oxygen in red and nitrogen in blue, are colored, respectively. Three sugars are shown for each hCEACAM1 molecule a, b and c (labelled in green, cyan and magenta, respectively). **b** Modeled N-linked sugars N-acetylglucosamine (NAG) and β-d-Mannose (BMA) onto the N70, N77, and N81 residues of human CEACAM1 molecules (a, b) do not appear to block any of the ABED face hydrogen-bond interactions shown. Modeled sugars are shown by stick diagrams and labeled as above. **c** Surface representation (green, cyan, and magenta) for molecules a, b, and c, respectively, and stick representation of modeled sugar molecules. GFCC' and ABED face interactions are depicted by black oblong and three modeled NNB sugars on each hCEACAM1 molecule residues N70, N77 and N81 do not appear to block ABED and GFCC' face interactions. NNB sugars and hCEACAM1 molecules are colored and labeled as above.

formed in both oligomers were mostly identical (Fig. 4b) with an (RMSD) of 0.6 Å (over 1536 atoms) and with similar GFCC'-face mediated interactions.

Our observation of a secondary interaction site in association with the ABED face as observed in the oligomeric structure provided an opportunity to better understand its role in the formation of higher-order oligomers. We therefore determined the crystallographic Debye-Waller factor (temperature factor or B factor) of all three molecules of the human CEACAM1 crystal structure reported here. This revealed higher thermal motion or dynamic mobility in the side-chains of some of the ABED interface- associated residues (Q27, Q53, Q54) (Fig. 5 and Supplementary Table 4). In addition, consistent with the notion that sugars exist as flexible molecules, we also observed that many of the potential sugar attachment site residues (N70, N77 and N81) also exhibited a high B factor (Fig. 5 and Supplementary Table 4).

Further, we found that the ABED-associated, hydrogen-bond interactions involve four glutamine residues (Q26, Q53, Q54 and Q76) that form a "four Q-pocket" (Supplementary Figs. 5b, 7a–b). The elevated B factors in some of these residues may also contribute to enhanced flexibility of the ABED face through the dual hydrogen-bond donor and acceptor properties of the side-chains associated with these residues (Supplementary Fig. 7a–b). Moreover, as we observed that the four Q-pocket together with the serine (S93) and asparagine (N23) residues which participate in the ABED face interactions are entirely conserved in human CEACAM family member IgV domains (Supplementary Fig. 5b), it is anticipated that such flexibility in the formation of oligomers through the ABED face may extend to other hCEACAM family

members. This is consistent with the observed structural flexibility revealed by previous molecular tomography studies which showed enough flexibility of the rat CEACAM1 IgV domain to be able to adopt several different conformations[23]. This observed flexibility is anticipated to be important in the formation of higher-order oligomers and micro-clusters where the GFCC' face serves as dominant initiating force due to its larger interface that includes extensive hydrogen-bond and hydrophobic interactions relative to that associated with the ABED face which is smaller in size with fewer potential interactions and thus serves as an auxiliary, flexible interface.

**A conserved metal ion binding site in hCEACAM1 and bridging by Ni$^{++}$ and Zn$^{++}$.** One of the important structural features of the hCEACAM1 GFCC' face is the involvement of the FG loop at the GFCC'-mediated dimeric interface, where residues such as Q89 of the F strand, E99 of the G strand and N97 within the FG loop form a hydrogen-bond network (Supplementary Fig. 1c). Of further importance to the structural integrity and conformation of the FG loop and G strand, we also observed that the main-chain nitrogen and carbonyl oxygen atoms of H105 make two hydrogen-bonds with the carbonyl oxygen of F9 and the main chain nitrogen atom of V11, respectively (Supplementary Fig. 8a). Moreover, our analysis of hCEACAM1 WT and V39A mutant structures (PDB code 2GK2, 6XNW)[7,8] and previously described hCEACAM6 WT crystal structures (PDB codes 4WHC, 4Y8A)[10] revealed evidence of hexadentate interactions between three H105 and three V106 hCEACAM1 residues contained within the G strands of the FG loop which coordinated

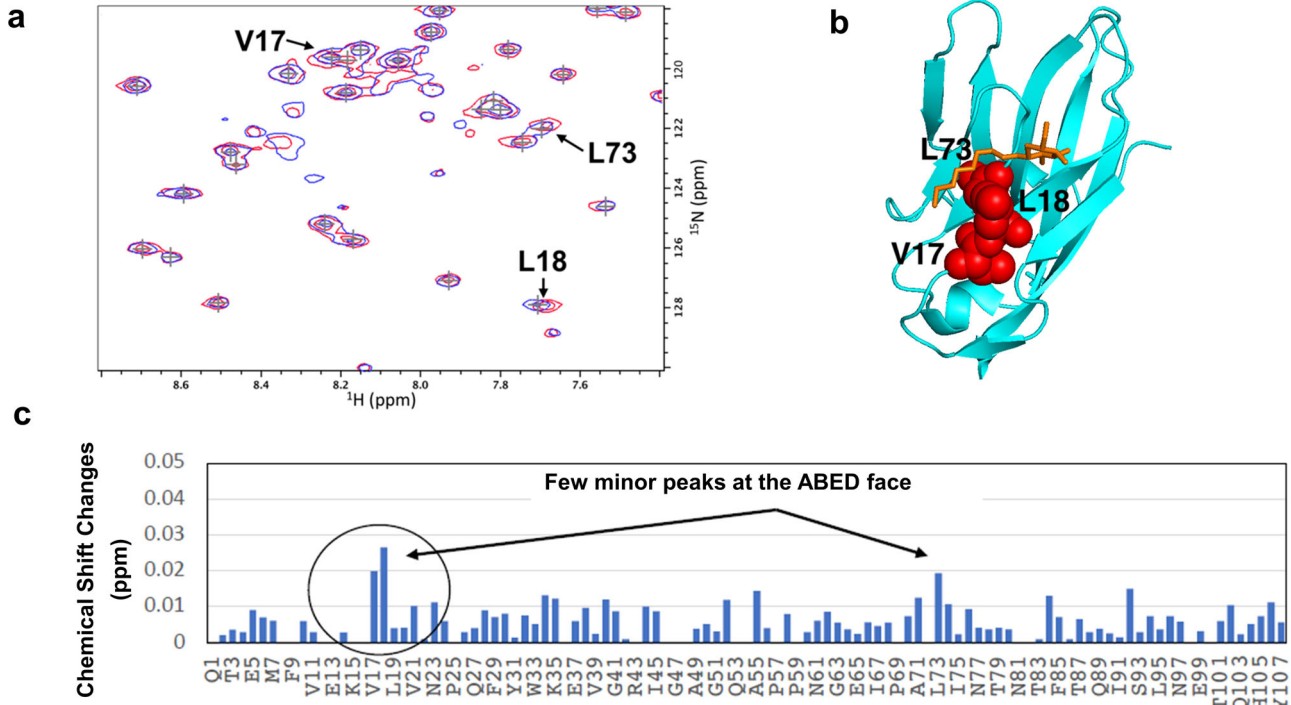

**Fig. 3 $^{15}$N-HSQC spectra of WT hCEACAM1 IgV domain binding with octyl beta-D-glucopyranoside (BOG). a** Wild type hCEACAM1 IgV domain (50 μM) binding with 10 mM octyl beta-D-glucopyranoside (BOG). An expanded region of overlaid $^{15}$N-HSQC spectra of hCEACAM1 WT alone (blue) and with BOG (red) reveal spectral changes for a few residues. Peaks shift observed for residues V17, L18 and L73 are shown by arrow. **b** Mapping of residues V17, L18 and L73 with the largest peak shift changes upon BOG binding onto the hCEACAM1 crystal structure (cyan) with bound BOG (PDB ID 4QXW). Bound BOG is shown by orange stick representation. Residues V17, L18 and L73 are shown by red sphere. **c** Upon binding of BOG with the hCEACAM1 IgV domain, only a few peaks from residues that localized to the ABED face show significant shift changes, consistent with binding of BOG near a hydrophobic patch of residues L73 and V17 as observed in the crystal structures (PDB codes 4QXW, 6XNT).

with Ni$^{++}$ in the hCEACAM1 structures (Supplementary Fig. 8b) or Zn$^{++}$ in the hCEACAM6 structure[10] (Supplementary Fig. 8c). This resulted in the bridging of three hCEACAM1 or three hCEACAM6 molecules in the crystal structures, respectively. Interestingly, the ~90% sequence identity observed between the hCEACAM1 and hCEACAM6 IgV domains, (Supplementary Fig. 5b) which mediates their interactions[21], includes conservation of the H105 and V106 residues[7].

These studies suggest that these divalent cations may also contribute to the formation of hCEACAM1 oligomers through interactions in the G strand of the FG loop. We therefore pursued UV-spectroscopy studies of hCEACAM1 with various metal ions (Fig. 6a, b). In this assay, various metals were titrated at a 1:1 or 2.5:1 molar ratio relative to a 100 μM concentration of hCEACAM1. We found that Zn$^{++}$ or Ni$^{++}$, but not Ca$^{++}$ or Mn$^{++}$ or Li$^{+}$ ions, showed solution opacity changes and concentration-dependent aggregation of hCEACAM1 with greater induction observed upon addition of Zn$^{++}$ (Fig. 6a, b). To confirm this, we next carried out NMR spectroscopy studies. $^{15}$N-labeled WT hCEACAM1 spectra showed loss of hCEACAM1 NMR peak intensities in the presence of varying concentrations of ZnCl$_2$ without the emergence of new peaks (Fig. 7a, b and Supplementary Fig. 9a–e). Further, we observed that the NMR samples turned from clear to translucent, to opaque or precipitated with increasing amounts of ZnCl$_2$ (from 20 μM to 250 μM) with complete reversal through chelation of the divalent cation through addition of 1 mM EDTA (Fig. 7c and Supplementary Fig. 9f). Similar solution opacity changes and loss of NMR spectra peaks were also observed for $^{15}$N-labeled WT hCEACAM1 in the presence of NiCl$_2$ (Fig. 7d, e). The NMR spectra peak loss caused by addition of Zn$^{++}$ or Ni$^{++}$ were

nearly uniform for all residues as a result of CEACAM1 aggregation and/or precipitation (Supplementary Fig. 10). Further, addition of 100 mM imidazole, an analog for histidine side-chains, to $^{15}$N-labeled WT hCEACAM1 in the presence of 130 μM NiCl$_2$ reversed the reduction in the peak intensities of hCEACAM1 observed during binding to the Ni$^{++}$ ions (Supplementary Fig. 11). Consistent with these results, mutation of H105 to alanine abrogated Zn$^{++}$ or Ni$^{++}$ -mediated aggregation of hCEACAM1 (Fig. 6c, d) and no cross-linking of Ni$^{++}$ was detected with hCEACAM1 under oxidative conditions[30] (Supplementary Fig. 12). These structural analyses together with spectroscopy and NMR studies support the direct involvement of Zn$^{++}$ or Ni$^{++}$ binding to H105 residue in the G strand of the FG loop of the hCEACAM1 IgV domain in the formation of oligomers and micro-clusters.

**Discussion**

In this study, we investigated the mechanisms involved in the formation of hCEACAM1 multimers. Our atomic level structural studies show that a single hCEACAM1 IgV domain can simultaneously interact with two other hCEACAM1 IgV domains along the dominant GFCC' and minor ABED face and thus helps to reconcile the presence of ABED (PDB code 2GK2) and GFCC' (PDB code 4QXW) face-mediated interactions as observed in previously reported crystal structures[8,9]. This is interesting as previous surface plasmon resonance studies of rat CEACAM1 IgV predicted the existence of two binding sites on the IgV domain consistent with low-resolution topography studies of rat CEACAM1 which show evidence of CEACAM1 trimers and higher order oligomers[23]. Our studies thus provide confirmation for this and mechanistic insights into this observation based upon

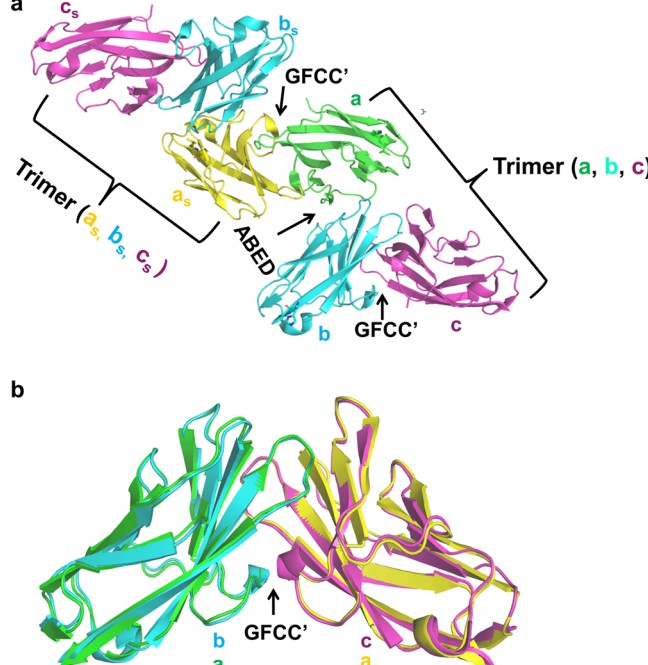

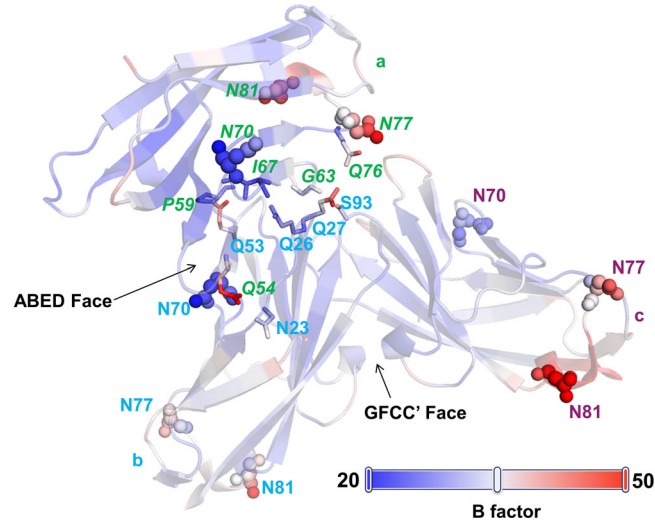

**Fig. 4 Human CEACAM1 higher-order oligomer formation is mediated by GFCC' face interactions between two oligomers. a** Human CEACAM1 higher-order oligomer formed by GFCC' face-mediated interactions between two hCEACAM1 oligomers as observed in the crystal structure. Human CEACAM1 oligomers present in the crystal asymmetric unit and symmetry-related oligomer are shown, wherein molecules a, b and c and its symmetry mates labeled $a_s$ (yellow), $b_s$ (cyan) $c_s$ (magenta) are shown by ribbon diagram. Central to higher-order oligomer formation is GFCC' face-mediated interactions between molecules a (green) from the first trimer (formed by molecules a, b and c) and its symmetry-mate $a_s$ (yellow) from the second trimer (formed by molecules $a_s$, $b_s$, and $c_s$). **b** Superimposition of the GFCC'-face mediated dimer present in both oligomers revealed similar GFCC-face mediated interactions, wherein GFCC' face dimers formed by molecules b and c, and by molecules a and it's symmetry-mate $a_s$ are shown by ribbon diagram.

**Fig. 5 Crystallographic Debye-Waller factor (temperature factor or B factor) assignment of the crystal structure of the hCEACAM1 IgV domain with GFCC' and ABED face interactions.** The ribbon diagram of the hCEACAM1 molecules (a, b, and c), glycosylation sites (N70, N77 and N81 residues shown by spheres), and interacting residues at ABED face (shown by sticks) with B-factor assignment. The loops, α helices, β strand interacting ABED face residues and glycosylation sites are colored based on B-factor range (blue-white-red, where blue minimum = 20, red maximum = 50). B-factor assignments revealed higher thermal motion or dynamic mobility in the side-chains (red) in some of the residues including Q27, Q53, Q54 at the ABED face and glycosylation site residues (N70, N77, and N81).

the atomic level resolution of a hCEACAM1 oligomer as reported here. Specifically, we observed that the GFCC'-based interface in this structure exhibited twice the area of the ABED interface. Consistent with this, the GFCC' interface involved 15 hydrogen-bond and 3 hydrophobic interactions as previously reported for a hCEACAM1 homodimeric IgV domain crystal structure (PDB code 4QXW)[9], compared to 6 hydrogen-bond and no hydrophobic interactions for the ABED interface. The ABED interface in the hCEACAM1 IgV oligomer also included four glutamine residues (four-Q pocket) which were conserved among other hCEACAM family members. This conservation of glutamine residues at this interface is expected to promote flexible hydrogen-bond interactions which might enable clustering of hCEACAM1 and other family members at the cell surface. We also observed subtle differences in the GFCC' interface in the context of an oligomer suggesting flexibility in this portion of the molecule that included an increase in the size of its interface area and interactions between the R38 and E37 residues that have not been previously observed in a homodimer.

As the structural studies reported here and previously involved a non-glycosylated N-domain that might interfere with multi-merization given their location within the IgV domain[8,26,27], we also sought to understand the effects of hCEACAM1 glycosylation on the ABED and GFCC' face interactions. To do so, we

modeled carbohydrate side-chain modifications and performed NMR spectroscopy that probed beta-octylglucoside (BOG) binding to hCEACAM1. This modeling and testing of our assumptions based upon the observed BOG interactions with hCEACAM1 predicted that the presence of sugar molecules at the hCEACAM1 glycosylation sites associated with N70, N77 and N81 would not interfere with the observed ABED and GFCC' face interactions. Further, B factor analysis of the hCEACAM1 oligomeric structure showed that the sites predicted to be associated with carbohydrate side-chain modifications (N70, N77, N81) as well as the amino acid residues along the ABED interface were characterized by high thermal motion and thus, flexibility. These results are further supported by a recent elegant study published at the time of this manuscript's submission which directly demonstrates by NMR and structural (PDB code 7MU8) obser-vations that dimer formation through hCEACAM1 GFCC' face interactions is observed in the presence of partially glycosylated hCEACAM1[31].

Finally, we examined the role of metals on hCEACAM1 mul-timerization as previous studies showed $Ni^{++}$ and $Zn^{++}$-mediated bridging of 3 hCEACAM1 molecules by coordination through histidine and valine residues in the G strands of the FG loops[7,8]. We noticed the presence of $Zn^{++}$ coordination of the same G strand residues in a previously reported structure of hCEACAM6[10]. As such, we also investigated the biochemical role of $Zn^{++}$ and $Ni^{++}$ in regulating hCEACAM1 multimerization using UV spectroscopy, NMR and site-directed mutagenesis studies. This revealed evidence of reversible $Zn^{++}$-and $Ni^{++}$-mediated oligomerization of hCEACAM1 based upon competition with EDTA or imidazole, a histidine residue mimetic. Note that we didn't observe any paramagnetic broad-ening of HSQC peaks with the addition of $NiCl_2$. This suggest that the residual NMR peaks were from un-ligated hCEACAM1,

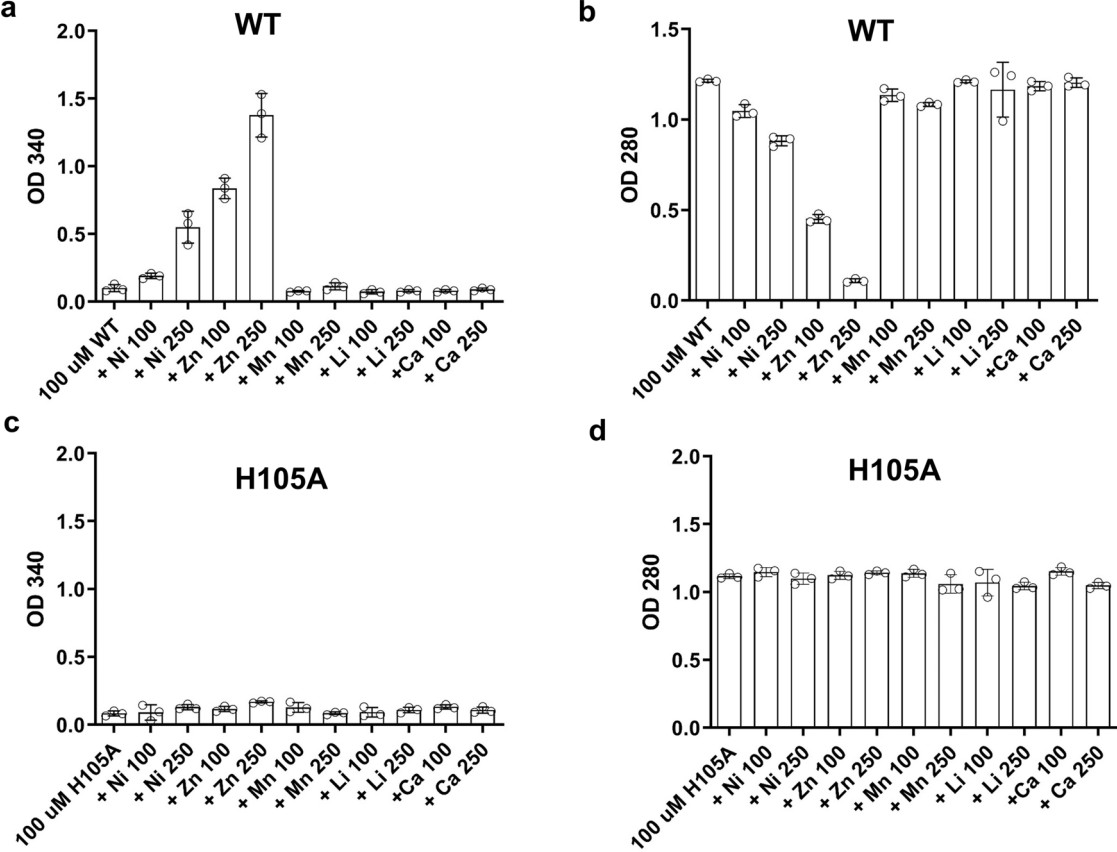

**Fig. 6 UV-spectroscopy studies of hCEACAM1 IgV WT and H105A mutant with various metal ions. a** Titration of hCEACAM1 WT 100 μM (with buffer) and with various metals ($Ni^{++}$, $Zn^{++}$, $Mn^{++}$, $Li^+$, $Ca^{++}$) at 1:1 and 1:2.5 ratio. OD analysis at 340 nm revealed $Zn^{++}$ or $Ni^{++}$ caused concentration dependent aggregation of the hCEACAM1 WT protein after 30-minute incubation. **b** OD analysis of supernatant after centrifugation at 280 nm for WT protein and with various metals ($Ni^{++}$, $Zn^{++}$, $Mn^{++}$, $Li^+$, $Ca^{++}$) at 1:1 and 1:2.5 ratio. **c** Titration of H105A mutant 100 μM (with buffer) and with various metals ($Ni^{++}$, $Zn^{++}$, $Mn^{++}$, $Li^+$, $Ca^{++}$) at 1:1 and 1:2.5 ratio. OD analysis at 340 nm revealed no $Zn^{++}$ or $Ni^{++}$ mediated aggregation observed for the H105A mutant protein. **d** OD analysis of H105A mutant supernatant after centrifugation at 280 nm alone and with various metals ($Ni^{++}$, $Zn^{++}$, $Mn^{++}$, $Li^+$, $Ca^{++}$) at 1:1 and 1:2.5 ratio. The mean values with standard deviations are shown in bar graph with error bars of the triplicate samples. Source data are provided in Supplementary Data 1.

and the metal bound hCEACAM1 molecules were likely in aggregates or precipitates. Interestingly at the basal state, CEACAM1 is known to form cis-dimers via a GXXXG motif (G432-6436) embedded in the transmembrane domain and inside-out $Ca^{++}$-dependent calmodulin binding downregulates cis-homodimerization of hCEACAM1 into monomers; these further facilitate trans-homophilic binding and downstream signaling[24,32]. Our studies thus implicate an additional mechanism that regulates oligomerization which is associated with a distinct binding motif in the G strand of the FG-loop of the IgV domain and involves specific metals in a concentration-dependent, dynamic process. Similar $Zn^{++}$ and $Ni^{++}$-mediated dynamic and reversible oligomerization in coordination through histidine residues was observed with APOBEC3G (A3G)[33], a DNA cytidine deaminase and with carbon monoxide dehydrogenase (CooJ)[34,35], respectively, using similar experimental approaches as reported here. In these studies, NMR titration experiments showed 50 mM $Zn^{++}$-dependent A3G (300 μM) oligomerization and transmission electron microscopy (TEM) showed 1 mM $Ni^{++}$-dependent CooJ (100 μM) oligomerization, which were reversible in both cases in the presence of EDTA[33–35].

Overall, our studies provide structural insights into how hCEACAM1 could exist as heterogeneous mixtures of oligomers and in diverse conformational states on the cell surface through primary GFCC' and secondary ABED interfaces that are mediated by the IgV domain which we show is able to simultaneously bind two hCEACAM1 molecules (Fig. 8). In this model, homo-dimerization through the GFCC' face enables highly flexible ABED-mediated interactions to form oligomers and higher-order oligomers. These are formed by symmetry mates of molecules a, b and c as observed in the crystal structure and are also based upon GFCC' face interactions which may be further facilitated by metal ions such as $Zn^{++}$ and $Ni^{++}$ (Fig. 8). It is not known how this model relates to cis or trans interactions or their functional implications, however. It is equally plausible that it represents a means to direct hCEACAM1 to an inactive state on the cell surface when in cis or to intracellular signal transduction when the GFCC' and consequently ABED interactions occur in trans across two cells. Nonetheless, our proposed model and the results of our biophysical and structural studies help to better understand previous observations and provide insights into the structural basis for the formation of hCEACAM1 oligomers (Supplementary Fig. 13) with implications for other hCEACAM family members. They may also potentially provide insights into the nature of the lower affinity heterophilic ligands for hCEACAM1 such as PD1[13] and TIM-3[9] which are likely to be highly influenced by the effects of avidity that would be facilitated by hCEACAM1 oligomerization.

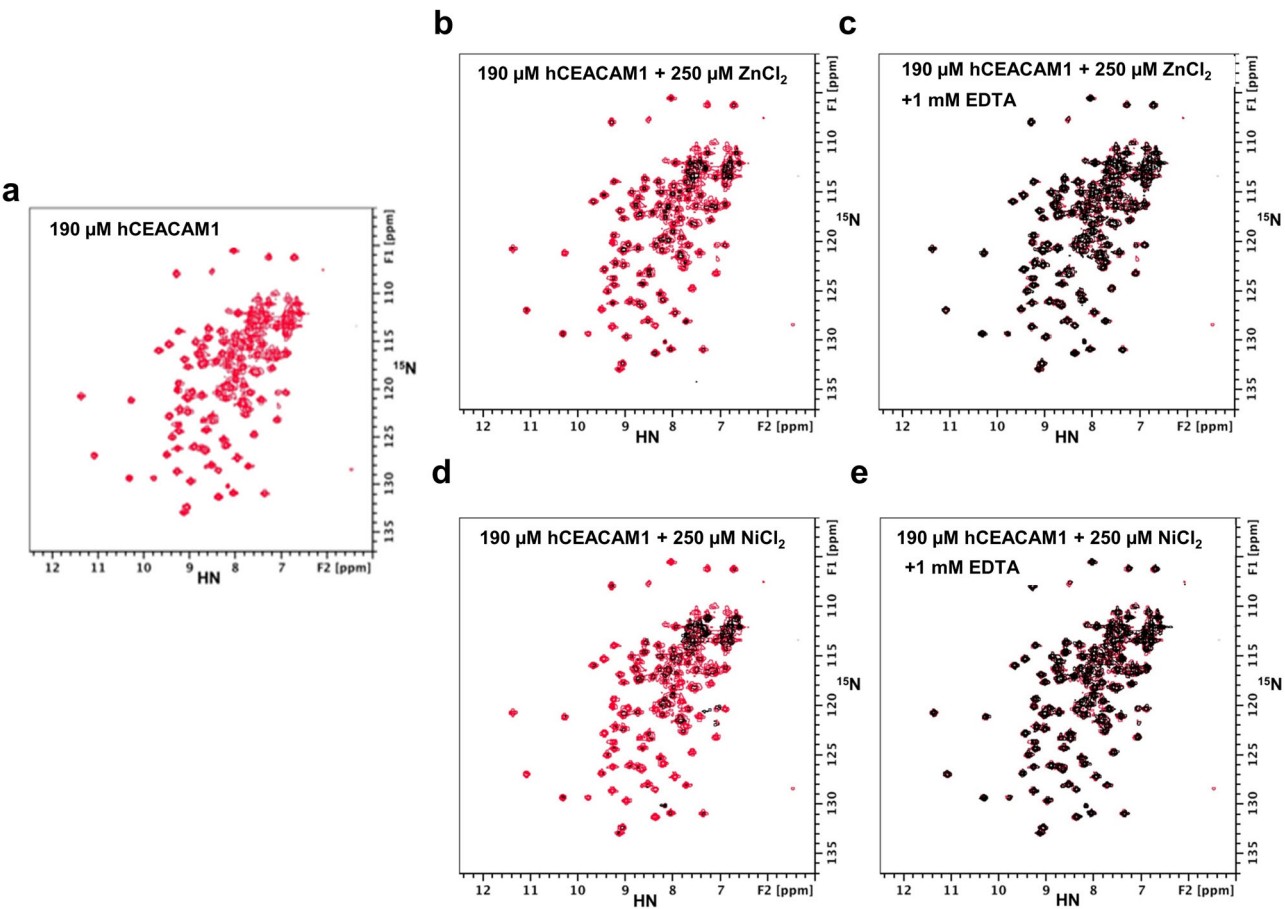

**Fig. 7 $^{15}$N-HSQC spectra of WT hCEACAM1 IgV domain with zinc chloride (ZnCl$_2$) and with nickel chloride (NiCl$_2$) and effect of EDTA.** $^{15}$N-HSQC spectra of 190 μM $^{15}$N-labeled WT hCEACAM1 IgV domain alone (panel **a**, peaks in red), and in the presence of 250 μM ZnCl$_2$ (panel **b**, weakened peaks in black), and both 250 μM ZnCl$_2$ and 1 mM EDTA (panel **c**, recovered peaks in black). The weakened and missing backbone amide peaks caused by zinc bridging induced oligomerization/aggregation as shown in middle **b** panel are recovered after addition of EDTA (right **c** panel) to sequester zinc ions. $^{15}$N-HSQC spectra of 190 μM $^{15}$N-labeled WT hCEACAM1 IgV domain in the presence of 250 μM NiCl$_2$ (panel **d**, weakened peaks in black), and both 250 μM NiCl$_2$ and 1 mM EDTA (panel **e**, recovered peaks in black). The weakened and missing backbone amide peaks caused by zinc bridging as shown in middle **b** panel or nickel bridging as shown in middle (**d**) panel are recovered after addition of EDTA to sequester zinc or nickel ions.

## Methods

**Protein expression, $^{15}$N labeling/culture preparation, refolding and purification.** hCEACAM1 IgV domain protein expression and purification were done mostly following our previously published protocols[7,9]. For unlabeled hCEACAM1 IgV protein, competent *E. coli* BL21 (DE3) were transformed with a pET21d vector containing human CEACAM1 IgV gene insert. Transformants were grown in 1 L of LB media supplemented with 100 μg/mL of ampicillin and induced with 1 mM IPTG after reaching an OD600nm of 0.8 at 37 °C. Next, the cultures were grown 4 more hours before harvesting cell pellets by centrifugation. For $^{15}$N-isotopic labeling of human CEACAM1 IgV, M9 minimal media containing 42 mM Na$_2$HPO$_4$, 22 mM KH$_2$PO$_4$, 8.6 mM NaCl, 0.1% $^{15}$N-NH$_4$Cl (Cambridge Isotope Laboratories), 0.4% glucose, 0.1 mM CaCl$_2$, 2 mM MgSO$_4$, and 1 μg/mL thiamine was used. Growth of $^{15}$N-hCEACAM1 IgV transformant was initially started in 5 mL of LB media with 100 μg/mL of ampicillin, which was further diluted 500-fold into 50 mL of M9 media plus 100 μg/mL ampicillin and grown overnight at 37 °C. Later, 50 mL grown culture was added to 450 mL M9 media and induced with 1 mM IPTG after reaching an OD600nm of 0.8 at 37 °C.

For refolding and purification of both unlabeled and $^{15}$N-labeled hCEACAM1 protein, the cell pellets containing expressed hCEACAM1 IgV protein were suspended in a resuspension buffer containing 20 mM Tris pH 7.5 and 500 mM NaCl, ruptured by sonication and centrifuged. The cell pellets were further washed in subsequent cycles of re-suspension, sonication, and centrifugation in resuspension buffer plus 0.2% Triton X-100 then in 10 mM Tris 8.0 and 1 M NaCl buffer and finally in 20 mM Tris pH 7.5 and 500 mM NaCl. The washed inclusion body pellet was dissolved in 5 ml Urea buffer containing 30 mM Tris pH 8.3, 8 M urea, and 150 mM NaCl and refolding was carried out with dropwise (~0.1 mL/min) addition of solubilized inclusion bodies into 250 ml of 50 mM CHES buffer pH 9.2 with 0.5 M L-arginine at 4 °C and stirred overnight.

Next, refolded unlabeled and $^{15}$N-labeled hCEACAM1 protein were concentrated and dialyzed against 4 liter of 10 mM Tris dialysis buffer pH 8.0 for 24 h and the dialysis buffer was changed 3 times. Dialyzed protein was filtered and then purified using MonoQ ion exchange column (GE Healthcare Life Sciences) using NaCl gradient of 0 mM to 200 mM in 10 mM Tris dialysis buffer pH 8.0. Peak fractions containing hCEACAM1 protein was verified using sodium dodecyl sulfate polyacrylamide gel electrophoresis (SDS-PAGE) under reducing conditions and further loaded onto a HiPrep 16/60 Sephacryl S-200 HR column (GE Healthcare Life Sciences) for size-exclusion chromatography in a buffer containing 10 mM HEPES, 150 mM NaCl pH 7.4 The final purity was >95%, as judged by SDS-PAGE.

**Crystallization and structure refinement of hCEACAM1 oligomer.** Cell pellets containing expressed hCEACAM1 IgV domain was purified using our previously published protocols[7,9]. Purified protein was concentrated to 15 mg/ml and after the various rounds of the crystallization screening, diffraction quality crystals were grown at 25 °C via sitting drop method in a reservoir solution containing 0.2 M Ammonium sulfate, 0.1 M HEPES pH 7.5, 25% w/v Polyethylene glycol 3350. The diffraction data was collected at the National Synchrotron Light Source (NSLS, Upton, NY, USA) beamline X25 and processed through iMOSFLM[36]. The structure of the hCEACAM1 oligomer in the C121 space group was determined using molecular replacement method using polyalanine model of our published hCEACAM1 homodimer crystal structure (PDB code 4QXW) as a search model and refined after various iterative rounds of simultaneous model building to final $R$ and $Rfree$ values of 18.4% and 23.7% using REFMAC[37] integrated to CCP4 suite[38] and COOT[39], respectively. Data collection and refinement statistics are listed in Table 1.

All the figures and comparison of the crystallographic Debye-Waller factor (temperature factor or B factor) of all three molecules of the hCEACAM1 oligomer

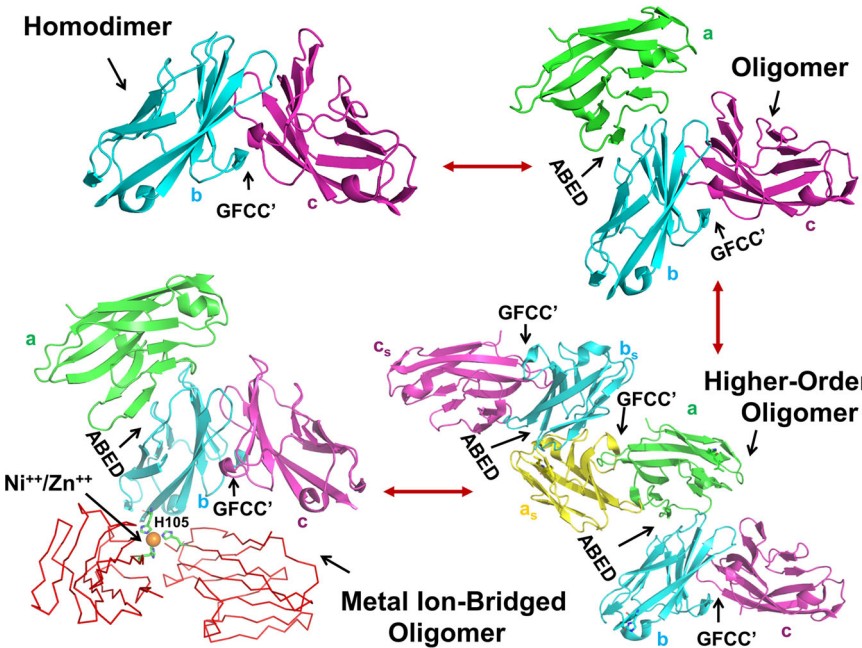

**Fig. 8 Oligomerization model of human CEACAM1.** Human CEACAM1 dimerization is mediated by primary GFCC' face which enables oligomerization through flexible ABED face as observed in the hCEACAM1 oligomeric crystal structure for molecules a (green), b (cyan) and c (magenta). The GFCC' face is a central force of this oligomeric assembly as it not only governs human CEACAM1 monomer–dimer equilibrium and interactions with ligands, but also enables highly flexible ABED-mediated interactions to form oligomers (wherein molecules b (cyan) and c (magenta) mediate GFCC' face interactions and molecules a (green) and b (cyan) mediate ABED face interactions) and higher-order oligomers (two hCEACAM1 oligomers as observed in the crystal structure, formed by molecules a, b and c and its symmetry mates labeled $a_s$ (yellow), $b_s$ (cyan) $c_s$ (magenta), wherein molecules a (green) and $a_s$ (yellow) mediate similar GFCC' face interactions observed for molecules b (cyan) and c (magenta)). Having flexibility in GFCC' and ABED face enables hCEACAM1 molecules to further bridge to other CEACAM1 molecules (shown here by red ribbon representation) through metal ions ($Ni^{++}$ or $Zn^{++}$, shown by orange sphere) via residues H105 as shown by green stick representation.

structure was done using PyMOL (DeLano Scientific). Sequence alignments of hCEACAM family members were done using Clustal Omega[40]. The residues level hydrogen-bond interactions and interface area as observed in the hCEACAM1 oligomer crystal structure for GFCC' and ABED face was determined using PDB PISA (proteins, interfaces, structures and assemblies)[41].

**Structural modelling and attachment of the N-linked sugars.** Based on the superimposition of mouse CEACAM1 crystal structure (PDB code 1L6Z) onto all three chains of hCEACAM1 oligomer using Pymol, wherein a conserved N70 glycosylation site residue was resolved in mouse CEACAM1 structure with two molecules of N- linked acetylglucosamine (NAG) NAG and a molecule of β-d-Mannose (BMA), two molecules of NAG and one molecule BMA were modeled on the hCEACAM1 N70 residue first for all three molecules of the hCEACAM1 oligomer. Then based on N70-linked sugar molecules, attachment of the N-linked sugars was performed for N77 and N81 residues of all three molecules of the hCEACAM1 oligomer using Coot. Comparison of the N-linked sugars hCEACAM1 oligomer model and WT homodimer with bound BOG (PDB code 4QXW) was performed using Pymol. In addition to manual modeling, sugar molecule modeling was also performed with the CHARMM glycosylation server[29].

*Mutagenesis, expression, and purification of H105A mutant.* Human CEACAM1 H105A mutagenesis studies were carried out by using the PCR-based QuikChange II Site-Directed Mutagenesis Kit (Agilent Technologies) with mutant forward primer sequence 5′-ctcgagttaatacacggcaaactgaccggttgcttcttcattaa-3′ and reverse primer sequence 5′-ttaatgaagaagcaaccggtcagtttgccgtgtattaactcgag-3′. The previously described pET21d plasmid containing human CEACAM1 IgV gene insert was used as the template for generating the H105A mutant. PCR reactions for introducing this H105A mutation were run for 16 cycles of 30 s at 95 °C and 1 min at 55 °C, followed by 6 min at 68 °C. The resulting hCEACAM1-H105A mutant plasmid was verified by standard Sanger DNA sequencing. For expression and purification of H105A mutant, similar protocols were used as applied for WT expression and purifications.

*UV-spectroscopy studies of hCEACAM1 WT and H105A with various metal ions.* In this assay, 100 μM concentration of tagless hCEACAM1 WT or H105A purified protein in 10 mM HEPES, 150 mM NaCl, pH 7.4 was incubated with various metals (ZnCl$_2$, NiCl$_2$, MnCl$_2$, LiCl, CaCl$_2$,) at a 1:1 or 2.5:1 molar ratio for 30 min.

First, OD measurement was performed at 340 nm to assess the aggregation and then after centrifugation for 2 min at 10,000 rpm, clear supernatant OD was also measured at 280 nm. In addition, blank experiments with buffer (10 mM HEPES, 150 mM NaCl, pH 7.4) were performed with or without various metals at the highest concentration used in this study (250 μM). This blank experiment didn't show any significant optical absorbance at 340 nm with all the metals.

*Oxidative cross-linking studies of hCEACAM1 with nickel.* Purified hCEACAM1 IgV (400 μM) was incubated with 250 uM NiCl$_2$ for 30 min in HEPES buffer (10 mM HEPES, 150 mM NaCl, pH 7.4). Crosslinking was performed by the addition of KHSO$_5$ (800 μM) for 30 min and then quenched by addition of equal volume of an SDS-PAGE denaturing buffer. Evaluation of crosslinking was performed by direct size assessment by SDS- PAGE electrophoresis.

*NMR spectroscopy studies of hCEACAM1 with various metal ions.* NMR samples of 190 μM $^{15}$N-labeled CEACAM1 were prepared with varying concentration of metal ion solutions in 20 mM HEPES, pH 7.4, 150 mM NaCl and 5% D$_2$O. The $^{15}$N-HSQC experiments were performed at 25 °C on a Bruker Avance II 600 MHz spectrometer equipped with a Prodigy Cryoprobe. The NMR data were acquired with 512 and 80 complex points in the direct HN and indirect $^{15}$N dimension respectively. NMR spectra were processed with Bruker Topspin software.

**Statistics and reproducibility**. The X-ray data and structure refinement statistics for the hCEACAM1 oligomer crystal structure is shown in Table 1. UV-spectroscopy studies were performed on triplicate samples ($n = 3$) for each metal titration experiment with hCEACAM1 as shown in in Fig. 6. The mean values with standard deviations are shown in bar graph with error bars.

**Reporting summary**. Further information on research design is available in the Nature Research Reporting Summary linked to this article.

## Data availability

The atomic coordinates and structure factors were deposited with RCSB accession code 7RPP. All relevant data are available upon request.

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

## Acknowledgements

The authors acknowledge thankful support from all the NSLS beamline scientists (Upton, NY, USA) for X-ray data collection and thanks various NMR facilties including HMS BIO-NMR (Boston, MA, USA), DFCI NMR core (Boston, MA, USA) facilities for NMR data collection. This work was supported by the NIH Grant 5R01DK051362-21 and the High Pointe Foundation to R.S.B.

## Author contributions

A.K.G., and Z.-Y.J.S. performed biophysical characterization, x-ray crystallography and NMR experiments. A.K.G. performed x-ray crystallization and structure determination. Z.-Y.J.S. carried out NMR experiments. Y.-H.H., W.M.K. and C.Y. performed analysis of hCEACAM1 oligomer. G.P. provided structural expertise in crystallization and structural comparison. R.S.B. established CEACAM1 project, and together with A.K.G., G.P., W.M.K., Z.-Y.J.S., Y.-H.H. and N.B. wrote the manuscript. R.S.B. and A.K.G are the senior author of this paper.

## Competing interests

The authors declare the following competing interests: R.S.B. has several issued and pending patents describing potential therapeutic strategies for regulating CEACAM1.
