## [Peer Review File · Communications Biology]

Reviewers' comments:

Reviewer #1 (Remarks to the Author):

In their manuscript entitled "Structural analysis of human CEACAM1 oligomerization", Gandhi et al. explore the molecular basis for higher-order oligomerization of human CEACAM1 (hCC1) that exceeds the established homodimer formation. CEACAM1 is known to engage into higher order complexes, such as dimers, trimers or microclusters that determine its effects on cellular regulation. For this purpose, the authors used a non-glycosylated recombinant IgV-hCC1 domain, which formed oligomers (trimers) that yielded crystals produced by the sitting drop method suitable for diffraction. Structure resolution was 2.2Å (PDB code: 7RPP).

The authors show that GFCC' - interface mediated oligomerization of the V domains conveyed exposure of a flexible ABED interface enabling further molecular engagement of the hCC1-IgV domains into trimers. This interaction proved to depend on divalent metal cations, such as Zn²⁺ or Ni²⁺, but not on the canonical Mn²⁺/Ca²⁺/Mg²⁺-dependent cluster coordination. In their study, the authors also sought to provide insight into hCC1 oligomer formation with regards to CEACAM1 IgV-resident N-linked glycosylation. For quite a while, it has been in the center intense whether homophilic or heterophilic interactions between CEACAMs or heteroligands depend on the heavy glycosylation decorating CEACAM1. In general, the N-terminal domain of CC1 contains 3 candidate Asn residues that could carry cognate glycans. With aid of overlays of glycan structures and the use of glycan mimetics (partially also published by the same group previously), and using extensive model comparisons as well as novel sugar molecule modeling, the authors could excluded that glycans impair domain interaction/clustering. This is corroborated by a recently published work of Belcher Dufrisne et al., Structure, 2022: 30, 1-13.

The authors present a concise, elegant and high-quality study that confers novel insight into domain organization and clustering of the extracellular N-terminal domain of hCC1. Previous reports detailed that the GFCC' interface is required to initiate molecular interactions between CEACAMs. However, the authors show that additional hydrogen bonds connecting ABED faces are involved in intermolecular clustering but appear to be yet dependent on interactions initiated via the GFCC' surface. They state that the N-terminal domain of hCC contains in fact two interfaces (GFCC' and ABED) that are involved in heterologous intermolecular interactions, with the ABED interface engaging in previously undescribed H-bonding between different monomers.

Furthermore, the functional relevance of the metal ion binding site for multimerization in the hCC1-IgV FG loop is demonstrated by UV spectroscopy / NMR and titration of ion concentrations. Interestingly, clusters can be dissolved by addition of imidazole or chelation of Ni²⁺ /Zn²⁺ by EDTA demonstrating the cluster formation is reversible under non-oxidizing conditions.

This well-designed study is presented in a succinct manuscript that provides extended and novel insight into the evolution of higher order clusters/oligomers of hCC1 simultaneously using both GFCC' and ABED interfaces that is dependent on divalent metal ions, but neither dependent nor impeded by N-linked IgV-resident glycans. In conjunction, this has not been demonstrated so far and provides important information on the organization of immune checkpoint inhibitor molecules. In the future, it will be imported to further explore the relevance of these data for CEACAM1 orientation and oligomerization on the cell surface and clarify whether Ni²⁺/Zn²⁺-mediated CEACAM1 clustering occurs in the context of pathophysiology. Also, the newly identified interfaces for intermolecular clustering may be exploited to manipulate CEACAM-interactions to influence immune reactions or signaling of CEACAM multimers in cancer cells.

However, I recommend to address the following points:

1. Regarding dimer and trimer formation – is there a known threshold concentration of recombinant

hCC1/IgV domain for multimerization, especially beyond homodimerization?

2. Nickel ions can oxidize (although they are not as prone to oxidation like iron) and crosslink proteins under certain conditions – see <https://pubs.acs.org/doi/pdf/10.1021/tx960170i>. Has this been considered especially in the context of cellular settings and oxidative environments? Have the authors considered titrating multimerization of hCC1-IgV in an oxidizing buffer? Is it at all possible that Ni crosslinks of CCs on the cell surface or in vitro play a role in signaling? Could this be of interest in settings like inflammation/tumor biology? If data is available, please include this in the manuscript

3. Could the authors comment on whether it is possible to show higher order complexes/oligomers >3 monomer units by density gradient centrifugation / crosslinks or other methods? Can the observed trimer act as a seed for initiating further oligomerization of complexes? If any information is available, please, include this information in the manuscript.

4. Can different multimers (2mer, 3mer, higher order oligomers /multimers) specifically identified by spectrometry? If so, please provide additional information in the figure/text.

5. The Methods are kept quite short, although the preparation of the protein has been described in previous publications, further details on the expression method/purification would be appreciated

6. Methods: please describe labeling /culture procedure of CEACAM1 with N15, for NMR

7. References: the formatting guidelines of the journal should be met, quite a few references are not cited correctly/miss page/volume information or DOIs

8. Discussion: The mechanism of Ca²⁺ ions/calmodulin binding to CC1 and subsequent inside-out dissociation of its dimers does not relate mechanistically to the CEACAM1-“chelation” mediated by metal ions on the outside of the cell. I would recommend to strengthen the differences between these observations. (line 371f)

9. Line 245: doubling of words “in the”

Reviewer #2 (Remarks to the Author):

This manuscript provides a structural analysis of how human CEACAM oligomerization may occur based on a combination of X-ray, NMR and modeling studies. The X-ray structures show that both GFCC' and ABED face interactions are possible, which suggests mechanisms for how higher order oligomeric structures may form. The authors take into consideration how glycosylation sites may affect oligomerization through modeling of these as well as NMR binding studies of carbohydrate mimics. They also investigate the role of metal ions in promoting oligomerization. In general the findings seem reasonable but not complete to the level needed for publication. A number of questions need to be addressed.

Major points:

1. The structural model of oligomerization should be tested, for example, by making mutations at either the GFCC' or ABED interfaces and then measuring the degree to which homodimerization or oligomerization occurs in solution. This would make for a more compelling case that the structures may have biological relevance.

2. In the binding studies with BOG, the chemical shift changes in Fig 3a are difficult to see and should be presented in a more expanded view of the 2D NMR spectrum. These changes should also be mapped onto the structure and compared with the modeled sugar positions. Also, the shifts are quite small, suggesting weak binding of BOG. What is the KD? Does the weak binding suggest that BOG is not a good model for sugar binding?

3. In Figure 7, the peak intensity loss should be quantified more clearly, particularly since the authors have the assignments. Which residues undergo loss of peak intensity? From Fig 7b, it seems that it is a subset of the residues. Since the authors have the assignments, they should be able to map peak intensity changes onto the structure to see if they are consistent with their oligomerization model.

Minor points:

4. The term "dynamic NMR studies" on line 126 is misleading and should be re-phrased.
5. In supplemental Figure 1 c, make the role of the FG loop clearer by highlighting or labeling it.

Reviewer #3 (Remarks to the Author):

This manuscript provides additional data (NMR and a new X-ray structure) that shed light on homotypic N-domain interactions that may facilitate the cell-surface oligomerization of CEACAM1. Dimerization through contacts on one face of the domain (GFCC') were well established in previous publications, and the possibility of ABDE face interactions were emphasized in the publication of the first crystal structure in 2006 and subsequent EM data in 2009. However, the new data shows some flexibility in the GFCC' interactions and a somewhat different set of interactions through the ABDE face. Moreover, the authors have modeled in glycoforms showing that at least the initial residues of native glycans can be accommodated in the ABDE mediated dimer, and they followed up on the potential importance of a Ni²⁺ binding site seen between three molecules in different unit cells with NMR and optical studies of ion induced aggregation. In short, significant new data is presented in support of the potential importance of high order oligomerization in cell-surface signaling by this molecule.

The presentation is a little lengthy with several figures devoted to NMR and optical data associated with the effect of metal ion binding and interactions with octyl-glucoside. The significance of these studies seems marginal. The ion binding studies are done at very high concentrations, well above anything likely to occur in vivo. Octyl glucoside certainly doesn't represent glycan interactions very well. More of this data could be moved to a supplement.

Some minor issues:

The authors should consider the possibility that some of the changes in optical absorbance come from formation of insoluble Zn or Ni hydroxides. I don't believe the authors give the pH of solutions studied; they should.

Lithium is a +1, not +2 ion.

Most complexes of Ni²⁺ are paramagnetic – could some of the NMR broadening in this case come from paramagnetic effects?

It is a little surprising that the loss of NMR peak intensity on adding Zn is not more uniform as one might expect for aggregation. The authors might comment.

The authors should alert the reader to the fact that their glycan modeling only includes the first 3 sugars of native glycans.

Dear Reviewers,

We thank the reviewers for their useful insights and supportive comments regarding our manuscript entitled “Structural analysis of human CEACAM1 oligomerization.” Below we address the reviewers’ concerns, comments, and suggestions. The reviewers’ comments below are in bold and our responses are in un-bolded text. We believe all the reviewers’ insights and comments have significantly improved the manuscript and included structural studies.

Reviewers' comments:

Reviewer #1 (Remarks to the Author):

In their manuscript entitled “Structural analysis of human CEACAM1 oligomerization”, Gandhi et al. explore the molecular basis for higher-order oligomerization of human CEACAM1 (hCC1) that exceeds the established homodimer formation. CEACAM1 is known to engage into higher order complexes, such as dimers, trimers or microclusters that determine its effects on cellular regulation. For this purpose, the authors used a non-glycosylated recombinant IgV-hCC1 domain, which formed oligomers (trimers) that yielded crystals produced by the sitting drop method suitable for diffraction. Structure resolution was 2.2Å (PDB code: 7RPP).

The authors show that GFCC´- interface mediated oligomerization of the V domains conveyed exposure of a flexible ABED interface enabling further molecular engagement of the hCC1-IgV domains into trimers. This interaction proofed to depend on divalent metal cations, such as Zn²⁺ or Ni²⁺, but not on the canonical Mn²⁺/Ca²⁺/Mg²⁺-dependent cluster coordination. In their study, the authors also sought to provide insight into hCC1 oligomer formation with regards to CEACAM1 IgV-resident N-linked glycosylation. For quite a while, it has been in the center intense whether homophilic or heterophilic interactions between CEACAMs or heteroligands depend on the heavy glycosylation decorating CEACAM1. In general, the N-terminal domain of CC1 contains 3 candidate Asn residues that could carry cognate glycans. With aid of overlays of glycan structures and the use of glycan mimetics (partially also published by the same group previously), and using extensive model comparisons as well as novel sugar molecule modeling, the authors could excluded that glycans impair domain interaction/clustering. This is corroborated by a recently published work of Belcher Dufrisne et al., Structure, 2022: 30, 1-13.

The authors present a concise, elegant and high-quality study that confers novel insight into domain organization and clustering of the extracellular N-terminal domain of hCC1. Previous reports detailed that the GFCC´ interface is required to initiate molecular interactions between CEACAMs. However, the authors show that additional hydrogen bonds connecting ABED faces are involved in intermolecular clustering but appear to be yet dependent on interactions initiated via the

GFCC' surface. They state that the N-terminal domain of hCC contains in fact two interfaces (GFCC' and ABED) that are involved in heterologous intermolecular interactions, with the ABED interface engaging in previously undescribed H-bonding between different monomers.

Furthermore, the functional relevance of the metal ion binding site for multimerization in the hCC1-IgV FG loop is demonstrated by UV spectroscopy / NMR and titration of ion concentrations. Interestingly, clusters can be dissolved by addition of imidazole or chelation of Ni²⁺ /Zn²⁺ by EDTA demonstrating the cluster formation is reversible under non-oxidizing conditions.

This well-designed study is presented in a succinct manuscript that provides extended and novel insight into the evolvement of higher order clusters/oligomers of hCC1 simultaneously using both GFCC' and ABED interfaces that is dependent on divalent metal ions, but neither dependent nor impeded by N-linked IgV-resident glycans. In conjunction, this has not been demonstrated so far and provides important information on the organization of immune checkpoint inhibitor molecules. In the future, it will be imported to further explore the relevance of these data for CEACAM1 orientation and oligomerization on the cell surface and clarify whether Ni²⁺/Zn²⁺-mediated CEACAM1 clustering occurs in the context of pathophysiology. Also, the newly identified interfaces for intermolecular clustering may be exploited to manipulate CEACAM-interactions to influence immune reactions or signaling of CEACAM multimers in cancer cells.

We thank the Reviewer #1 for very inspiring feedback of our structural studies in deciphering the basis of hCEACAM1 oligomerization. We are pleased that the reviewer understands the importance of our structural information in human CEACAM1 signaling and the therapeutic targeting of cancer.

However, I recommend addressing the following points:

1. Regarding dimer and trimer formation – is there a known threshold concentration of recombinant hCC1/IgV domain for multimerization, especially beyond homodimerization?

The reviewer asked a very important question. The reported *dissociation constant* (K_D) for hCEACAM1 IgV dimerization is approximately 450 nM (Bonsor et al, 2015) however higher order dissociation constants are unknown; previous studies (Korotkova et al, 2008) observed higher order species in their AUC experiments but without clear measurements on concentrations thresholds or calculatable dissociation constants of higher order oligomerization. Based on our experience in crystallization trials, CEACAM1 IgV homodimerization was observed in the crystal structure determination (PDB ID 4QXW) at concentrations approaching 425 μ M concentration; the higher order oligomer crystal structure (PDB ID 7RPP) was resolved at concentrations approaching 1.2 mM. However, crystallization does not provide a reliable determination of the exact or accurate threshold concentration for higher-order oligomerization.

2. Nickel ions can oxidize (although they are not as prone to oxidation like iron) and crosslink proteins under certain conditions – see <https://pubs.acs.org/doi/pdf/10.1021/tx960170i>. Has this been considered especially in the context of cellular settings and oxidative environments? Have the authors considered titrating multimerization of hCC1-IgV in an oxidizing buffer?

We thank the reviewer for these important suggestions.

- As suggested by the reviewer, we considered the possibility of nickel oxidation and protein crosslinking in oxidative conditions and carried out oxidative cross-linking experiments with hCEACAM1 the presence of nickel using protocols referred in the study by the reviewer. However, we didn't observe any evidence of nickel-mediated crosslinking of hCEACAM1 as shown using SDS-page electrophoresis below (**Supplemental Fig. 12**). We have updated the manuscript method section with these details.

Supplementary Fig. 12. Oxidative cross-linking studies of hCEACAM1 with nickel: SDS-PAGE showing no changes in hCEACAM1 molecular weight across various lanes either in 400 μ M hCEACAM1-IgV with 250 μ M NiCl_2 (lane 1) or in 400 μ M hCEACAM1-IgV with 250 μ M NiCl_2 and 800 μ M KHSO_5 (lane 2). Molecular weight marker is shown in lane a.

“Oxidative cross-linking studies of hCEACAM1 with nickel: Purified hCEACAM1 IgV (400 μ M) was incubated with 250 μ M NiCl_2 for 30 minutes in HEPES buffer (10 mM HEPES, 150 mM NaCl, pH 7.4). Crosslinking was performed by the addition of KHSO_5 (800 μ M) for 30 minutes and then quenched by addition of equal volume of an SDS-PAGE

denaturing buffer. Evaluation of crosslinking was performed by direct size assessment by SDS-PAGE electrophoresis.”

- Further, we have performed site-directed mutagenesis studies with purified mutant hCEACAM1-H105A protein to confirm a role of H105 in metal-mediated hCEACAM1 oligomerization. We observed that once the H105 residue is mutated to an alanine, hCEACAM1 IgV oligomerization is mostly abrogated in the presence of divalent metal cations (Ni^{++} or Zn^{++}). The revised figures Fig.6c and Fig.6d depict UV-spectroscopy studies of the hCEACAM1-H105A mutant in the presence of various metal ions, wherein titration of hCEACAM1-H105A mutant (100 μM with buffer) and with various metals (Ni^{++} , Zn^{++} , Mn^{++} , Li^+ , Ca^{++}) at 1:1 and 1:2.5 ratio showed no metal-mediated oligomerization as observed for WT (Fig.6a and Fig.6b). Thus, the H105A mutagenesis studies validate the direct role of residue H105 on hCEACAM1 on coordination with Zn^{++} or Ni^{++} in metal-mediated oligomerization of hCEACAM1. We have updated the manuscript text between lines “328-333” and method section with these details.

“Consistent with these results, mutation of H105 to alanine abrogated Zn^{++} or Ni^{++} - mediated aggregation of hCEACAM1 (**Fig. 6c-d**) and no cross-linking of Ni^{++} was detected with hCEACAM1 under oxidative conditions³⁰ (**Supplemental Fig. 12**). These structural analyses together with spectroscopy and NMR studies support the direct involvement of Zn^{++} or Ni^{++} binding to H105 residue in the FG loop of the hCEACAM1 IgV domain in the formation of oligomers and micro-clusters.”

“Mutagenesis, expression, and purification of H105A mutant: Human CEACAM1 H105A mutagenesis studies were carried out by using the PCR-based QuikChange II Site-Directed Mutagenesis Kit (Agilent Technologies) with mutant forward primer sequence 5'-ctcgagttaatacacggcaactgaccggttgcttcttcattaa-3' and reverse primer sequence 5'-ttaatgaagaagcaaccggtcagtttgccgtgtattaactcgag-3'. The previously described pET21d plasmid containing human CEACAM1 IgV gene insert was used as the template for generating the H105A mutant. PCR reactions for introducing this H105A mutation were run for 16 cycles of 30 s at 95°C and 1 min at 55°C, followed by 6 min at 68°C. The resulting hCEACAM1-H105A mutant plasmid was verified by standard Sanger DNA sequencing. For expression and purification of H105A mutant, similar protocols were used as applied for WT expression and purifications.”

Fig. 6 UV-spectroscopy studies of hCEACAM1 IgV WT and H105A mutant with various metal ions. a Titration of hCEACAM1 WT 100 μ M (with buffer) and with various metals (Ni^{2+} , Zn^{2+} , Mn^{2+} , Li^+ , Ca^{2+}) at 1:1 and 1:2.5 ratio. OD analysis at 340 nm revealed Zn^{2+} or Ni^{2+} caused concentration dependent aggregation of the hCEACAM1 WT protein after 30-minute incubation. **b** OD analysis of supernatant after centrifugation for 2 min at 10000 rpm at 280 nm for WT protein and with various metals (Ni^{2+} , Zn^{2+} , Mn^{2+} , Li^+ , Ca^{2+}) at 1:1 and 1:2.5 ratio. **c** Titration of H105A mutant 100 μ M (with buffer) and with various metals (Ni^{2+} , Zn^{2+} , Mn^{2+} , Li^+ , Ca^{2+}) at 1:1 and 1:2.5 ratio. OD analysis at 340 nm revealed no Zn^{2+} or Ni^{2+} mediated aggregation observed for the H105A mutant protein. **d** OD analysis of H105A mutant supernatant after centrifugation for 2 min at 10000 rpm at 280 nm alone and with various metals (Ni^{2+} , Zn^{2+} , Mn^{2+} , Li^+ , Ca^{2+}) at 1:1 and 1:2.5 ratio.

Is it at all possible that Ni crosslinks of CCs on the cell surface or in vitro play a role in signaling? Could this be of interest in settings like inflammation/tumor biology? If data is available, please include this in the manuscript.

The reviewer raises a very important point. Although we did not observe crosslinking by nickel under oxidative conditions, a role of nickel or zinc in hCEACAM1 signaling cannot be ruled out. So far, various crystal structures of hCEACAM1 WT or mutant (PDB code 2GK2, 6XNW) and hCEACAM6 (PDB codes 4WHC, 4Y8A) shows coordination by Ni^{2+}

or Zn⁺⁺ in the hCEACAM1 or hCEACAM6 structures and our UV-spectroscopy and NMR studies of hCEACAM1 with these metal ions confirm metal-mediated oligomerization. Future studies are certainly needed to address these questions and role of metal-mediated oligomerization of hCEACAM1 in inflammation/tumor biology.

3. Could the authors comment on whether it is possible to show higher order complexes/oligomers >3 monomer units by density gradient centrifugation / crosslinks or other methods?

The reviewer asked a series of very interesting questions, and we appreciate the reviewer's insights.

- Regarding the possibility of showing higher order hCEACAM1 oligomerization by other methods, It is promising that hCEACAM1 displayed similar characteristics of higher order oligomeric forms in an analytical ultracentrifugation experiment, although data was not shown, in an excellent structural study performed by Korotkova et al. in 2008 (Korotkova, N., Yang, Y., Le Trong, I., Cota, E., Demeler, B., Marchant, J., Thomas, W. E., Stenkamp, R. E., Moseley, S. L. & Matthews, S. Binding of Dr adhesins of Escherichia coli to carcinoembryonic antigen triggers receptor dissociation. *Mol Microbiol* 67, 420-434, doi:10.1111/j.1365-2958.2007.06054.x (2008)).
- Similarly, low resolution (20 Å) molecular tomography studies of liposome-immobilized glycosylated rat CEACAM1 protein by Klaile, E. et al. in 2009 (Klaile, E. et al. The CEACAM1 N-terminal Ig domain mediates cis- and trans-binding and is essential for allosteric rearrangements of CEACAM1 microclusters. *Journal of Cell Biology* (2009) doi:10.1083/jcb.200904149) revealed various higher order states of CEACAM1 oligomerization that included micro-clusters (> 3 mer) of closely associated molecules. Results of tomography studies were further confirmed by cross-linking studies revealing stabilization of dimers, trimers, tetramers, and higher multimers states. Thus, results of these previously published studies confirm the possibility of showing CEACAM1 higher-order oligomerization states by various methods. We have included these observations for the reviewer's benefits below and revised manuscript text between lines "79-84" to read as follows.

"In support of hCEACAM1 oligomerization, cross-linking, and low resolution (20 Å) molecular tomography studies of liposomal-immobilized rat CEACAM1 (IgV and 3 IgC2 domains), sharing ~43% sequence identity with hCEACAM1 in the IgV domain, revealed multiple states of CEACAM1 that included monomers, dimers, trimers and micro-clusters of closely associated molecules²³."

Fig. a Previously published molecular tomography of rat CEACAM1 D (1–4) showing trimer consisting of a monomer (lime green) binding via its D1 domain to an A-dimer (red/blue). **b** Molecular tomography of CEACAM1 ectodomains attached to liposomes showing various states of CEACAM1. **c** Purified rat CEACAM1 D (1–4) were cross-linked with BS3 in the presence or absence of 2.5 mM Ca⁺⁺, 2.5 mM Mg⁺⁺, 3 mM EDTA, and Ni-NTA liposomes at a high protein/lipid ratio (hp; 1:10 [wt/wt]) or a low protein/lipid ratio (lp; 1:90 [wt/wt]) in various combinations. Addition of Ca/Mg ions or EGTA did not significantly change the proportion of the different crosslinked species, but EDTA decreased the abundance of crosslinked trimers/tetramers/multimers.

Can the observed trimer act as a seed for initiating further oligomerization of complexes? If any information is available, please, include this information in the manuscript.

We agree with the reviewer. It is entirely possible that a trimer could act as a seed in the higher-order oligomerization process via GFCC' face interactions between various trimers. Consistent with the reviewer insights, we observed a possible role of trimer formation as a seed for higher-order oligomerization in revised **Fig. 4a**, which shows formation of a higher-order oligomer as the GFCC' face of one trimer interacts with the GFCC' face of a second trimer as observed in the oligomer crystal structure.

Fig. 4a Human CEACAM1 higher-order oligomer formation is mediated by GFCC' face interactions between two oligomers. a Human CEACAM1 higher-order oligomer formed by GFCC' face-mediated interactions between two hCEACAM1 oligomers as observed in the crystal structure. Human CEACAM1 oligomers present in the crystal

asymmetric unit and symmetry-related oligomer are shown, wherein molecules a, b and c and its symmetry mates labeled a_s (yellow), b_s (cyan) c_s (magenta) are shown by ribbon diagram. Central to higher-order oligomer formation is GFCC' face-mediated interactions between molecules a (green) from the first trimer (formed by molecules a, b and c) and its symmetry-mate a_s (yellow) from the second trimer (formed by molecules a_s , b_s , and c_s).

4. Can different multimers (2mer, 3mer, higher order oligomers /multimers) specifically identified by spectrometry? If so, please provide additional information in the figure/text.

The reviewer’s insights are consistent with our thinking and in our previous publication, we pursued size-exclusion chromatography with multi-angle light scattering (SEC-MALS) for purified samples of hCEACAM1 Ig-V and various mutants (V39A, I91A, N97A, E99A) for detection of various states and monodispersity. In these (SEC-MALS) experiments, we identified monomeric (N97A), transition state (V39A) and dimeric states (WT, I91A, E99A) at 100 μ M concentrations. Thus, we agree with the reviewer regarding use of SEC-MALS to detect various oligomeric states of hCEACAM1 at experimental concentrations. We have modified the text in the manuscript between lines “72-76” to add this information.

“Consistent with the importance of the GFCC’ face for ligand interactions, the GFCC’ face also governs various states involved in hCEACAM1 dimerization. As such, size-exclusion chromatography with multi-angle light scattering (SEC-MALS) studies of hCEACAM1 Ig-V and GFCC’-face mutants (V39A, I91A, N97A, E99A) detected various states of hCEACAM1 including monomeric, transition and dimeric states⁷.”

We have added our results of (SEC-MALS) studies from our previous publication below for the reviewer’s benefit (Gandhi, A. K. *et al.* Structural basis of the dynamic human CEACAM1 monomer-dimer equilibrium. *Communications Biology* 4, (2021).)

5. The Methods are kept quite short, although the preparation of the protein has been described in previous publications, further details on the expression method/purification would be appreciated.

6. Methods: please describe labeling /culture procedure of CEACAM1 with N15, for NMR.

We thank the reviewer for these important suggestions. We have described all the following details of protein expression, the labeling/culture protocol and the purifications in the revised manuscript method section as shown below.

“Protein Expression, ¹⁵N labeling/culture preparation, refolding and purification.

hCEACAM1 IgV domain protein expression and purification were done mostly following our previously published protocols^{8,10}. For unlabeled hCEACAM1 IgV protein, competent *E. coli* BL21 (DE3) were transformed with a pET21d vector containing human CEACAM1 IgV gene insert. Transformants were grown in 1 L of LB media supplemented with 100 µg/mL of ampicillin and induced with 1 mM IPTG after reaching an OD_{600nm} of 0.8 at 37°C. Next, the cultures were grown 4 more hours before harvesting cell pellets by centrifugation. For ¹⁵N-isotopic labeling of human CEACAM1 IgV, M9 minimal media containing 42 mM Na₂HPO₄, 22 mM KH₂PO₄, 8.6 mM NaCl, 0.1% ¹⁵N-NH₄Cl (Cambridge Isotope Laboratories), 0.4% glucose, 0.1 mM CaCl₂, 2 mM MgSO₄, and 1 µg/mL thiamine was used. Growth of ¹⁵N-hCEACAM1 IgV transformant was initially started in 5 mL of LB media with 100 µg/mL of ampicillin, which was further diluted 500-fold into 50 mL of M9 media plus 100 µg/mL ampicillin and grown overnight at 37°C. Later, 50 mL grown culture was added to 450 mL M9 media and induced with 1 mM IPTG after reaching an OD_{600nm} of 0.8 at 37°C.

For refolding and purification of both unlabeled and ¹⁵N-labeled hCEACAM1 protein, the cell pellets containing expressed hCEACAM1 IgV protein were suspended in a resuspension buffer containing 20 mM Tris pH 7.5 and 500 mM NaCl, ruptured by sonication and centrifuged. The cell pellets were further washed in subsequent cycles of re-suspension, sonication, and centrifugation in resuspension buffer plus 0.2% Triton X-100 then in 10 mM Tris 8.0 and 1 M NaCl buffer and finally in 20 mM Tris pH 7.5 and 500 mM NaCl. The washed inclusion body pellet was dissolved in 5 ml Urea buffer containing 30 mM Tris pH 8.3, 8 M urea, and 150 mM NaCl and refolding was carried out with dropwise (~0.1 mL/min) addition of solubilized inclusion bodies into 250 ml of 50 mM CHES buffer pH 9.2 with 0.5 M L-arginine at 4°C and stirred overnight.

Next, refolded unlabeled and ¹⁵N-labeled hCEACAM1 protein were concentrated and dialyzed against 4 liter of 10 mM Tris dialysis buffer pH 8.0 for 24 hours and the dialysis buffer was changed 3 times. Dialyzed protein was filtered and then purified using MonoQ ion exchange column (GE Healthcare Life Sciences) using NaCl gradient of 0 mM to 200 mM in 10 mM Tris dialysis buffer pH 8.0. Peak fractions containing hCEACAM1 protein was verified using sodium dodecyl sulfate polyacrylamide gel electrophoresis (SDS-PAGE) under reducing conditions and further loaded onto a HiPrep 16/60 Sephacryl S-200 HR column (GE Healthcare Life Sciences) for size-exclusion chromatography in a

buffer containing 10 mM HEPES, 150 mM NaCl pH 7.4 The final purity was >95%, as judged by SDS-PAGE.”

7. References: the formatting guidelines of the journal should be met, quite a few references are not cited correctly/miss page/volume information or DOIs.

We apologize to the reviewer for our oversight. We have reformatted all the references as per the journal guidelines and added all the missing information.

8. Discussion: The mechanism of Ca²⁺ ions/calmodulin binding to CC1 and subsequent inside-out dissociation of its dimers does not relate mechanistically to the CEACAM1-“chelation” mediated by metal ions on the outside of the cell. I would recommend to strengthen the differences between these observations. (line 371f)

We thank the reviewer. We have strengthened the differences between Ca⁺⁺-mediated calmodulin binding and “CEACAM1-chelation” mediated by these metal ions. We have modified the text in the manuscript between lines “388-395” to add this information.

“Interestingly at the basal state, CEACAM1 is known to form cis-dimers via a GXXXG motif (G432-6436) embedded in the transmembrane domain and inside-out Ca⁺⁺-dependent calmodulin binding downregulates cis-homodimerization of hCEACAM1 into monomers; these further facilitate trans-homophilic binding and downstream signaling^{24,32}. Our studies thus implicate an additional mechanism that regulates oligomerization which is associated with a distinct binding motif in the FG-loop of the IgV domain and involves specific metals in a concentration-dependent, dynamic process.”

9. Line 245: doubling of words “in the”

We thank the reviewer. We have removed doubling of the words in the revised manuscript.

Reviewer #2 (Remarks to the Author):

This manuscript provides a structural analysis of how human CEACAM oligomerization may occur based on a combination of X-ray, NMR and modeling studies. The X-ray structures show that both GFCC' and ABED face interactions are possible, which suggests mechanisms for how higher order oligomeric structures may form. The authors take into consideration how glycosylation sites may affect oligomerization through modeling of these as well as NMR binding studies of carbohydrate mimics. They also investigate the role of metal ions in promoting oligomerization. In general the findings seem reasonable but not complete to the level needed for publication. A number of questions need to be addressed.

We thank the Reviewer #2 for encouraging feedback of our structural studies and pleased to see the reviewer finds results of these studies reasonable.

Major points:

1. The structural model of oligomerization should be tested, for example, by making mutations at either the GFCC' or ABED interfaces and then measuring the degree to which homodimerization or oligomerization occurs in solution. This would make for a more compelling case that the structures may have biological relevance.

We thank the reviewer for their very useful insight, and we agree with reviewer regarding use of mutagenesis and in solution studies to further confirm the role of GFCC' and ABED faces in human CEACAM1 oligomerization. To address these reviewer's concerns, we have revised the manuscript and supplemental figure to highlight GFCC' face N97A mutant which lacks GFCC' face-mediated dimerization with minor contacts in ABED face, and further pursued H015A mutant studies to provide clarity in role of the H105 residue in metal-mediated oligomerization. These changes include:

- We have revised the manuscript text and supplemental **Fig.3b** to highlight the mutation of GFCC' face residue N97 to alanine that results in abrogation of GFCC'-face mediated dimerization with only minor contacts through ABED face residues I67: Q26 as observed in the N97A mutant crystal structure (PDB ID 6XO1). Further, NMR spectroscopy studies of the N97A mutant in solution discussed in our previous publication showed that N97A mutation leads to shifting of the monomer–dimer equilibrium towards a monomeric form with an estimated molecular weight of 11 kD. Given the role of the GFCC' face as a major interface as observed the oligomer crystal structure with N97 residues making multiple hydrogen-bonded interactions with other GFCC' face residues including S32 and Q44 (**Supplemental Fig. 2a**), we expect the N97A mutation will lead to abrogation of a hCEACAM1 oligomerization state as well, thus highlighting the importance of the GFCC' face in oligomerization. To reflect these observations, we have updated the manuscript text between lines “248-252 to read as below.

- “In addition, the N97A mutation which disables GFCC'-mediated dimerization (PDB code 6XO1) supports the importance of the GFCC' face in oligomerization as well as this mutant does not show proper ABED face-associations and only exhibits minor contacts via hydrogen-bond interactions involving residues Q26 and I67 (**Supplemental Fig. 3b**)⁷.”

- Further, we have performed site-directed mutagenesis studies with purified mutant hCEACAM1-H105A protein to confirm a role of H105 in metal-mediated hCEACAM1 oligomerization. We observed that once the H105 residue is mutated to an alanine, hCEACAM1 IgV oligomerization is mostly abrogated in the presence of divalent metal cations (Ni^{++} or Zn^{++}). The revised figures Fig.6c and Fig.6d depict UV-spectroscopy studies of the hCEACAM1-H105A mutant in the presence of various metal ions, wherein titration of hCEACAM1-H105A mutant (100 μM with buffer) and with various metals (Ni^{++} , Zn^{++} , Mn^{++} , Li^+ , Ca^{++}) at 1:1 and 1:2.5 ratio showed no metal-mediated oligomerization as observed for WT (Fig.6a and Fig.6b). Thus, the H105A mutagenesis studies validate the direct role of residue H105 on hCEACAM1 on coordination with Zn^{++} or Ni^{++} in metal-mediated oligomerization of hCEACAM1. We have updated the manuscript text between lines “328-333” and method section with these details.

“Consistent with these results, mutation of H105 to alanine abrogated Zn^{++} or Ni^{++} - mediated aggregation of hCEACAM1 (**Fig. 6c-d**) and no cross-linking of Ni^{++} was detected with hCEACAM1 under oxidative conditions³⁰ (**Supplemental Fig. 12**). These structural analyses together with spectroscopy and NMR studies support the direct

involvement of Zn⁺⁺ or Ni⁺⁺ binding to H105 residue in the FG loop of the hCEACAM1 IgV domain in the formation of oligomers and micro-clusters.”

“Mutagenesis, expression and purification of H105A mutant: Human CEACAM1 H105A mutagenesis studies were carried out by using PCR-based QuikChange II Site-Directed Mutagenesis Kit (Agilent Technologies) with mutant forward primer sequence 5'-ctcgagttaatacacggcaaaactgaccggttgcttcttcattaa-3' and reverse primer sequence 5'-ttaatgaagaagcaaccggtcagttgcccgtgtattaactcgag-3'. The previously described pET21d plasmid containing human CEACAM1 IgV gene insert was used as the template for generating the H105A mutant. PCR reactions for introducing this H105A mutation were run for 16 cycles of 30 s at 95°C and 1 min at 55°C, followed by 6 min at 68°C. The resulting hCEACAM1-H105A mutant plasmid was verified by Sanger DNA sequencing. For expression and purification of H105A mutant, similar protocols were used as applied for WT expression and purifications.”

Fig. 6 UV-spectroscopy studies of hCEACAM1 IgV WT and H105A mutant with various metal ions. a Titration of hCEACAM1 WT 100 μM (with buffer) and with various metals (Ni⁺⁺, Zn⁺⁺, Mn⁺⁺, Li⁺, Ca⁺⁺) at 1:1 and 1:2.5 ratio. OD analysis at 340 nm revealed

Zn⁺⁺ or Ni⁺⁺ caused concentration dependent aggregation of the hCEACAM1 WT protein after 30-minute incubation. **b** OD analysis of supernatant after centrifugation for 2 min at 10000 rpm at 280 nm for WT protein and with various metals (Ni⁺⁺, Zn⁺⁺, Mn⁺⁺, Li⁺, Ca⁺⁺) at 1:1 and 1:2.5 ratio. **c** Titration of H105A mutant 100 μM (with buffer) and with various metals (Ni⁺⁺, Zn⁺⁺, Mn⁺⁺, Li⁺, Ca⁺⁺) at 1:1 and 1:2.5 ratio. OD analysis at 340 nm revealed no Zn⁺⁺ or Ni⁺⁺ mediated aggregation observed for the H105A mutant protein. **d** OD analysis of H105A mutant supernatant after centrifugation for 2 min at 10000 rpm at 280 nm alone and with various metals (Ni⁺⁺, Zn⁺⁺, Mn⁺⁺, Li⁺, Ca⁺⁺) at 1:1 and 1:2.5 ratio.

Thus, these studies demonstrate the role of GFCC' face residues in hCEACAM1 oligomerization and we hope that these additions address the reviewer's concern. It will be desirable for us to pursue additional ABED-face mutagenesis studies as suggested by the reviewer in the future.

2. In the binding studies with BOG, the chemical shift changes in Fig 3a are difficult to see and should be presented in a more expanded view of the 2D NMR spectrum. These changes should also be mapped onto the structure and compared with the modeled sugar positions. Also, the shifts are quite small, suggesting weak binding of BOG. What is the KD? Does the weak binding suggest that BOG is not a good model for sugar binding?

We appreciate the reviewer's suggestions as they allow us to apply these important details in hCEACAM1: BOG binding studies. To address reviewer's concerns, we have implemented the following changes and revised **Fig. 3**.

- As suggested by the reviewer, we have expanded the NMR spectral region as shown in revised **Fig. 3a** that contains peak shift changes as observed for residues V17, L18 and L73 and moved the full-size NMR spectrum to supplementary data as revised **Supplementary Fig. 6**.
- Also, according to the reviewer's suggestion, we have mapped the amino acid residues most affected by BOG binding, namely V17, L18 and L73, onto the crystal structure with bound BOG (PDB ID 4QXW) in a new **Fig. 3b** panel.
- The NMR peak shifts are too small even at 10mM BOG concentration, so we didn't attempt to measure a kD value for BOG binding. We agree with the reviewer that BOG is perhaps not the very best glycan mimic model for sugar binding, but overall support binding of BOG as observed in various hCEACAM1 crystal structures (PDB codes 4QXW, 6XNT). In addition, BOG binding overlaps with the glycan binding sites as shown in crystal structure of mouse CEACAM1 (PDB code 1L6Z) and recently resolved elegant crystal structure of partially glycosylated human CEACAM1 (PDB code 7MU8).

Fig. 3 ^{15}N -HSQC spectra of WT hCEACAM1 IgV domain binding with octyl beta-D-glucopyranoside (BOG). **a** Wild type hCEACAM1 IgV domain (50 μM) binding with 10mM octyl beta-D-glucopyranoside (BOG). An expanded region of overlaid ^{15}N -HSQC spectra of hCEACAM1 WT alone (blue) and with BOG (red) reveal spectral changes for a few residues. Peaks shift observed for residues V17, L18 and L73 are shown by arrow. **B** Mapping of residues V17, L18 and L73 with the largest peak shift changes upon BOG binding onto the hCEACAM1 crystal structure (cyan) with bound BOG (PDB ID 4QXW). Bound BOG is shown by orange stick representation. Residues V17, L18 and L73 are shown by red sphere. **c** Upon binding of BOG with the hCEACAM1 IgV domain, only a few peaks from residues that localized to the ABED face show significant shift changes, consistent with binding of BOG near a hydrophobic patch of residues L73 and V17 as observed in the crystal structures (PDB codes 4QXW, 6XNT).

3. In Figure 7, the peak intensity loss should be quantified more clearly, particularly since the authors have the assignments. Which residues undergo loss of peak intensity? From Fig 7b, it seems that it is a subset of the residues. Since the authors have the assignments, they should be able to map peak intensity changes onto the structure to see if they are consistent with their oligomerization model.

The NMR signal loss caused by addition of either nickel or zinc is nearly uniform for all residues as a result of CEACAM1 aggregation and/or precipitation. The HSQC spectra in **Fig. 7** are plotted on the same absolute scale, so that only the strongest NMR peaks of

the protein sample with metal ion are visible, leading to the impression of non-uniform intensity loss. Revised supplementary **Fig. 9** below plots the relative peak intensities of CEACAM1 residues with/without 65 μM NiCl_2 or 65 μM ZnCl_2 showing similar intensities (a few residues with higher values may be caused by error of overlapping peaks). We have revised the manuscript between lines “323-325” with these details.

“The NMR spectra peak loss caused by addition of Zn^{++} or Ni^{++} were nearly uniform for all residues, as a result of CEACAM1 aggregation and/or precipitation (**Supplemental Fig. 10**).”

Supplementary Fig. 10. ^{15}N -HSQC spectra of hCEACAM1 WT IgV with zinc chloride (ZnCl_2) or nickel chloride (NiCl_2) showing NMR signal loss caused by addition of nickel or zinc is nearly uniform of all residues. **a** Relative peak intensity of CEACAM1 residues with/without 65 μM ZnCl_2 . **b** Relative peak intensity of CEACAM1 residues with/without 65 μM NiCl_2 .

Minor points:

4. The term “dynamic NMR studies” on line 126 is misleading and should be re-phrased.

We apologize to the reviewer for this confusion. We have replaced term “dynamic NMR studies” on line 131 with “NMR studies” in the revised manuscript.

5. In supplemental Figure 1c, make the role of the FG loop clearer by highlighting or labeling it.

We thank the reviewer for this suggestion. We have updated the labelling in supplemental **Fig. 1c** to highlight the role of the FG and CC' loops.

Supplemental Fig.1c Snapshot of the GFCC' face interactions with electron density as observed between molecule *b* (cyan) and molecule *c* (magenta) residues in the crystal structure, wherein molecule *b* residues F29, Y34, V39, Q89, I91, N97 (cyan) make various hydrogen-bond and hydrophobic interactions with molecule *a* residues F29, Y34, V39, Q89, I91, and N97 (magenta). Residues are shown by sticks with corresponding electron density ($2F_o - F_c$ map at 1.0σ level). The carbon atoms in cyan (molecule *b*) or magenta (molecule *a*), carbonyl oxygen in red and nitrogen in blue, are colored respectively. CC' and FG loops are labeled accordingly.

Reviewer #3 (Remarks to the Author):

This manuscript provides additional data (NMR and a new X-ray structure) that shed light on homotypic N-domain interactions that may facilitate the cell-surface oligomerization of CEACAM1. Dimerization through contacts on one face of the domain (GFCC') were well established in previous publications, and the possibility of ABDE face interactions were emphasized in the publication of the first crystal structure in 2006 and subsequent EM data in 2009. However, the new data shows some flexibility in the GFCC' interactions and a somewhat different set of interactions through the ABDE face. Moreover, the authors have modeled in glycoforms showing that at least the initial residues of native glycans can be accommodated in the ABDE mediated dimer, and they followed up on the potential importance of a Ni²⁺ binding site seen between three molecules in different unit cells with NMR and optical studies of ion induced aggregation. In short, significant new data is presented in support of the potential importance of high order oligomerization in cell-surface signaling by this molecule.

The presentation is a little lengthy with several figures devoted to NMR and optical data associated with the effect of metal ion binding and interactions with octyl-glucoside. The significance of these studies seems marginal. The ion binding studies are done at very high concentrations, well above anything likely to occur in vivo. Octyl glucoside certainly doesn't represent glycan interactions very well. More of this data could be moved to a supplement.

We are pleased as Reviewer #3 finds our work of significant interest. We apologize to reviewer for lengthy presentation of the figures. To address the reviewer's concerns and avoid any superfluous details, we have simplified the main and supplemental figures with following changes.

- We have combined NMR ¹⁵N-HSQC studies of hCEACAM1 with zinc chloride and nickel chloride as a single combined figure (**Fig. 7**) and most of ¹⁵N-HSQC spectra of WT hCEACAM1 IgV domain with zinc chloride (ZnCl₂) and with nickel chloride (NiCl₂) are now presented in **Supplemental Figures. 8-12**.
- As suggested by the reviewer, we have moved the full-size NMR spectrum showing ¹⁵N-HSQC spectra of WT hCEACAM1 binding with BOG to supplementary data as supplemental **Fig. 6**. Hopefully to the reviewer's satisfaction, the revised **Fig. 3** shows NMR binding of BOG to hCEACAM1 in a more succinct manner.

Fig. 7 ^{15}N -HSQC spectra of WT hCEACAM1 IgV domain with zinc chloride (ZnCl_2) and with nickel chloride (NiCl_2) and effect of EDTA. ^{15}N -HSQC spectra of 190 μM ^{15}N -labeled WT hCEACAM1 IgV domain alone (panel a, peaks in red), and in the presence of 250 μM ZnCl_2 (panel b, weakened peaks in black), and both 250 μM ZnCl_2 and 1mM EDTA (panel c, recovered peaks in black). The weakened and missing backbone amide peaks caused by zinc bridging induced oligomerization/aggregation as shown in middle b panel are recovered after addition of EDTA (right c panel) to sequester zinc ions. ^{15}N -HSQC spectra of 190 μM ^{15}N -labeled WT hCEACAM1 IgV domain in the presence of 250 μM NiCl_2 (panel d, weakened peaks in black), and both 250 μM NiCl_2 and 1mM EDTA (panel e, recovered peaks in black). The weakened and missing backbone amide peaks caused by zinc bridging as shown in middle b panel or nickel bridging as shown in middle d panel are recovered after addition of EDTA to sequester zinc or nickel ions.

Fig. 3 ^{15}N -HSQC spectra of WT hCEACAM1 IgV domain binding with octyl beta-D-glucopyranoside (BOG). **a** Wild type hCEACAM1 IgV domain (50 μM) binding with 10mM octyl beta-D-glucopyranoside (BOG). An expanded region of overlaid ^{15}N -HSQC spectra of hCEACAM1 WT alone (blue) and with BOG (red) reveal spectral changes for a few residues. Peaks shift observed for residues V17, L18 and L73 are shown by arrow. **B** Mapping of residues V17, L18 and L73 with the largest peak shift changes upon BOG binding onto the hCEACAM1 crystal structure (cyan) with bound BOG (PDB ID 4QXW). Bound BOG is shown by orange stick representation. Residues V17, L18 and L73 are shown by red sphere. **c** Upon binding of BOG with the hCEACAM1 IgV domain, only a few peaks from residues that localized to the ABED face show significant shift changes, consistent with binding of BOG near a hydrophobic patch of residues L73 and V17 as observed in the crystal structures (PDB codes 4QXW, 6XNT).

Some minor issues:

The authors should consider the possibility that some of the changes in optical absorbance come from formation of insoluble Zn or Ni hydroxides. I don't believe the authors give the pH of solutions studied; they should.

We thank the reviewer for these important suggestions, and we have updated UV-spectroscopy methods with the pH of the solutions used. Further, as suggested by the reviewer, to address the possible formation of insoluble Zn or Ni hydroxides with any correlation in optical absorbance, we have performed a blank experiment with buffer (10

mM HEPES, 150 mM NaCl, pH 7.4) and with various metals at the highest concentration used in this study (250 μ M). This blank experiment didn't show any significant optical absorbance at 340 nm with all the metals. We have added these details in the methods section to read as follows.

The revised method read as follows.

“UV-spectroscopy studies of hCEACAM1 WT and H105A with various metal ions:

In this assay, 100 μ M concentration of tagless hCEACAM1 WT or H105A purified protein in 10 mM HEPES, 150 mM NaCl, pH 7.4 was incubated with various metals (ZnCl₂, NiCl₂, MnCl₂, LiCl, CaCl₂,) at a 1:1 or 2.5:1 molar ratio for 30 min. First, OD measurement was performed at 340 nm to assess the aggregation and then after centrifugation for 2 min at 10,000 rpm, clear supernatant OD was also measured at 280 nm. In addition, blank experiments with buffer (10 mM HEPES, 150 mM NaCl, pH 7.4) were performed with or without various metals at the highest concentration used in this study (250 μ M). This blank experiment didn't show any significant optical absorbance at 340 nm with all the metals.”

Lithium is a +1, not +2 ion.

We apologize to the reviewer for this oversight.

Most complexes of Ni²⁺ are paramagnetic – could some of the NMR broadening in this case come from paramagnetic effects?

It is a little surprising that the loss of NMR peak intensity on adding Zn is not more uniform as one might expect for aggregation. The authors might comment.

We didn't observe paramagnetic broadening after adding NiCl₂. We believe the NMR peaks shown in the HSQC spectrum are from un-ligated CEACAM1, and the Ni-bound CEACAM1 are in aggregate and/or precipitation states. It is likely that any exchange between the two species is slower and unobservable at the NMR time scale. We did, however, observe localized paramagnetic broadening of CEACAM1 residues after adding MnCl₂ (data not shown), unrelated to any aggregation/precipitation phenomenon, and likely caused by non-specific binding of Mn⁺⁺ ion due to charge-charge interactions.

The NMR signal loss caused by addition of nickel or zinc is nearly uniform for all residues, as a result of CEACAM1 aggregation and/or precipitation. The HSQC spectra the reviewer probably referred to are plotted on the same absolute scale, so that only the strongest NMR peaks of the protein sample with metal ion are visible, leading to the impression of non-uniform intensity loss. A revised supplemental **Fig. 10** below plots the relative peak intensities of CEACAM1 residues with/without 65 μ M ZnCl₂, showing uniform intensity loss.

Supplementary Fig. 10. ^{15}N -HSQC spectra of hCEACAM1 WT IgV with zinc chloride (ZnCl_2) or nickel chloride (NiCl_2) showing NMR signal loss caused by addition of nickel or zinc is nearly uniform of all residues. **a** Relative peak intensity of CEACAM1 residues with/without 65 μM ZnCl_2 . **b** Relative peak intensity of CEACAM1 residues with/without 65 μM NiCl_2 .

The authors should alert the reader to the fact that their glycan modeling only includes the first 3 sugars of native glycans.

We thank the reviewer for this suggestion. We have revised the text in the manuscript to read as follows on line 210: “This manual modeling of the first 3 sugars of the native glycans was based on the mouse CEACAM1 crystal structure (PDB code 1L6Z) wherein a conserved N70 glycosylation site residue was resolved and shown to be linked to two molecules of NAG and a molecule of BMA (**Supplemental Fig. 4a-b**)²⁹.”

REVIEWERS' COMMENTS:

Reviewer #2 (Remarks to the Author):

The authors have addressed all of my concerns adequately. The manuscript is now acceptable in my opinion.

Reviewer #3 (Remarks to the Author):

The revised version of this manuscript has satisfactorily addressed most the questions raised in the previous version. The results of several control experiments have been added, Figure 3 has been improved and some less important material has been moved to the supplement. The suggested lack of effects of native glycans on interactions through the ABDE face are still supported only by modeling and the presence of a BOG molecule in the crystal structure. The authors have more fairly presented their modeling arguments by point our that only the first three sugars of a typical N-glycan were modeled. However, it would have been useful to also point out that native N-glycans typically carry 7-12 sugars. Nevertheless, the manuscript does make a case for the potential importance of higher order interactions of CEACAM1 through the ABDE interface and metal coordination, and this is sufficiently well presented for publication.